# Adapting Actively on the Fly: Relevance-Guided Online Meta-Learning with Latent Concepts for Geospatial Discovery

## Abstract

In many real-world settings, such as environmental monitoring, disaster response, or public health, where data is costly and difficult to collect, strategically sampling from unobserved regions is essential for uncovering hidden risks or targets under tight resource constraints. Moreover, real-world geospatial data is often sparse, noisy, and geographically skewed, rendering most existing learning-based methods impractical. To address this, we propose a unified geospatial discovery framework that integrates active learning, online meta-learning, and concept-guided reasoning to support efficient decision-making under extreme data scarcity. Our approach introduces two key innovations: a *relevance-uncertainty guided sampling strategy* that uses structured relevance vectors based on domain-specific concepts (e.g., land cover, industrial proximity), enabling interpretable and adaptive sample selection; and a *relevance-aware meta-batch formation strategy* that promotes semantic diversity during online-meta updates, improving generalization in dynamic environments. Our experiments include actively searching for a specific land cover type under sparse training conditions and a strict sampling budget, as well as identifying cancer-causing PFAS (Per- and polyfluoroalkyl substances) contamination hotspots in U.S. surface water bodies, *a critical, real-world public health and environmental problem*. Despite limited observations and significant landscape shifts, our method reliably uncovers target land covers and contamination zones and adapts across space and time, showcasing its scalability, robustness, and potential to accelerate discovery in data-limited, high-stakes environments.

## 1 Introduction

Discovering targets of interest in large, costly-to-sample spaces is a core challenge across environmental monitoring, disaster response, public health, and other geospatial tasks, where targets of interest may include pollution hotspots, damaged regions, or regions at risk of disease, for example (Bondi et al., 2022). These domains often have a strict acquisition budget and limited previous observations, and must continue to make sampling decisions over time. Furthermore, tasks in environmental settings may involve diffuse phenomena that are inherently invisible in remote sensing imagery (Ayush et al., 2020). Decision-making in such environments must therefore be guided by uncertainty and sparse prior data, with a delicate balance between exploration (gathering new information), and exploitation (focusing on areas with the highest likelihood of targets).

Prior geospatial search methods (Sarkar et al., 2023a; 2024b) often rely on Reinforcement Learning (RL) to develop search policies that efficiently balance exploration and exploitation. However, these RL-based approaches often require millions of interaction steps - even in toy domains like Atari (Lattimore et al., 2013; Zhang et al., 2023; Schulman et al., 2017) - while in our setting, we operate under a strict budget of only ∼100 interaction steps, which is typical for real-world tasks like environmental monitoring, where labels are costly and each sample requires field effort (US EPA, 2018). This makes such RL methods impractical for real-world tasks. Partially observable Markov decision processes (POMDPs) are data-hungry as well, limiting their applicability (Kaelbling et al., 1998; Zheng et al., 2022; Ross et al., 2008). Classical bandit algorithms also offer an efficient way

to balance exploration and exploitation, but they lack mechanisms to exploit spatial or semantic structure in high-dimensional data (Li et al., 2010; Bubeck & Cesa-Bianchi, 2012). Such assumptions break down in complex settings like geospatial search, where observations are structured and highly correlated. Active and online learning techniques hold relevance, yet the dynamic nature of target states, such as pollution or disease, poses substantial challenges. Conventional meta-learning methods (Finn et al., 2017; 2018; 2019), designed to rapidly adapt policies, cannot be directly applied since we don't have the luxury of a predefined meta-training set. Furthermore, geospatial search in open-world settings imposes severe constraints; storing all observed data is infeasible, and identifying high-utility samples requires novel adaptive strategies. We formalize these challenges through our newly proposed *Open-World Learning for Geospatial Prediction and Sampling (OWL-GPS)* problem setting.

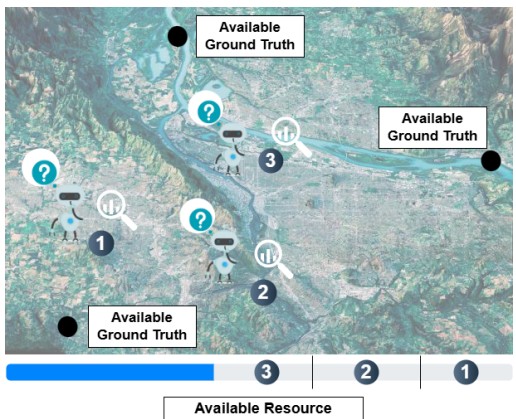

Figure 1: An illustration of the challenge of training a model with sparse and expensive ground truth data under resource constraints, a typical scenario in real-world environmental monitoring. As each query depletes available resources, it highlights the need for an intelligent policy that efficiently learns from limited data. The goal is to optimize data collection for continual model learning and maximal target discovery.

---

**OWL-GPS Assumptions**

(**OWL-GPS**) operates under three key constraints: (**i**) inputs arrive sequentially and must be actively selected and acted upon by the learned policy under a non-stationary distribution, (**ii**) once observed, samples cannot be revisited or replayed beyond a fixed *lifespan* due to strict memory constraints (see Sec. 4), and (**iii**) sampling budgets are strictly limited both during training and inference. These constraints reflect practical realities in environmental monitoring pipelines, and fundamentally diverge from assumptions in standard active learning (e.g., static unlabeled pools) and lifelong learning (e.g., task boundaries, replay buffers). OWL-GPS thus demands a new class of adaptive, non-revisitable, and policy-driven learning strategies.

---

To deploy models at a global scale with limited data, two core challenges arise: (1) selecting queries that both maximize information gain and target likely hotspots, and (2) learning adaptively as new observations arrive sequentially and distributions evolve. This requires a dynamic strategy that updates both what to sample and how to learn from it in real time.

Our key contributions are summarized as follows:

- We define the **Open-World Learning for Geospatial Prediction and Sampling (OWL-GPS)** setting, marked by policy-driven sampling, strict acquisition budgets, evolving distributions, and non-revisitable inputs, diverging sharply from assumptions in traditional active and lifelong learning.
- We propose a **concept-guided relevance encoder** that utilizes a Conditional Variational Auto-Encoder (CVAE) conditioned on domain concepts (e.g., land cover, facility proximity) to learn structured relevance vectors, enabling interpretable prediction and sampling.
- We develop a **relevance-aware meta-training strategy** that forms diverse, high-utility meta-batches on the fly via uncertainty and conceptual dissimilarity, unlike prior static-buffer or episodic approaches, and better suited for spatially continuous, non-stationary settings.
- We validate our framework on two **real-world discovery tasks**, *PFAS hotspot detection* and *rare land cover identification*, demonstrating robust performance under sparse supervision and distribution shift, setting a new OWL-GPS benchmark.

.

## 2 PROBLEM FORMULATION

Driven by real-world challenges, we propose the novel problem setting termed *OWL-GPS*. OWL-GPS captures the need to balance constrained data collection during training with effective decision-making at deployment. Specifically, we consider a scenario where, during training, we operate under a limited budget to acquire labeled data, aiming to learn a model that generalizes well across diverse geospatial regions. At inference time, a separate query budget is available; the objective expands to not only improving the model but also efficiently identifying regions with target presence.

This distinction introduces a core challenge: how to select queries during inference that both respect the query budget and maximize discovery of the target class. This is further exacerbated in real-world applications, where data distributions may differ significantly across regions due to domain shifts in environmental, spatial, or temporal factors. We formalize this as follows. Let $X_{train} = \{x_{train}^{(1)}, \ldots, x_{train}^{(N)}\}$ and $X_{test} = \{x_{test}^{(1)}, \ldots, x_{test}^{(M)}\}$ denote two sets of images covering diverse geospatial regions. Typically, $M > N$, as training samples are usually gathered from localized regions before the model is deployed in an open-world setting. Each image $x^{(j)}$ is composed of visual features that capture spatial and environmental factors. The $k$-th pixel in $x^{(j)}$, denoted $x_k^{(j)}$, corresponds to a binary label $y_k^{(j)} \in \{0, 1\}$ indicating the presence or absence of the target.

At each query step $t$, a policy $\pi_{\theta_{t-1}}$ selects a region $q_t$ from the unobserved set to query. After querying, the true labels $y^{(q_t)}$ are revealed for the selected region $x^{(q_t)}$, and predictions are made by the model using the current parameters $\theta_{t-1}$. The objective is to select a sequence of queries that maximizes the number of correctly identified target pixels (true positives) across all queried regions, subject to a total cost budget $\mathcal{C}$.

$$\max_{\{\pi_{\theta t-1}\}} \quad \sum_t \sum_{i=1}^{P} y_i^{(q_t(\pi_{\theta t-1}))} \cdot \mathbf{1}\left[ y_i^{(q_t(\pi_{\theta t-1}))} = \underbrace{\left[\pi_{\theta t-1}\left(x^{(q_t(\pi_{\theta t-1}))}\right)\right]_i}_{\text{i-th pixel predicted by the model given region } q_t(\pi_{\theta t-1})} \right]$$

$$\text{s.t.} \quad \sum_{t \geq 0} c\left(q_{t-1}(\pi_{\theta t-2}), q_t(\pi_{\theta t-1})\right) \leq \mathcal{C} \tag{1}$$

Here $c(i, j)$ is the cost associated with uncovering the ground-truth level $y^{(j)}$ associated with $x^{(j)}$ when initiating the query process from the region associated with $x^{(i)}$, and $P$ is the total number of pixels in a given region $x$. The indicator function ensures that only the correct target detections, corresponding to the binary label 1, are counted as success. The constraint enforces that the cumulative query cost does not exceed the total allowed budget. Intuitively, this objective captures the dual challenge at the heart of OWL-GPS: making the best use of a limited number of queries to discover target regions, despite potential domain shifts and training-data limitations. To the best of our knowledge, the OWL-GPS problem setting we propose is novel, thereby necessitating the development of a novel framework specifically designed to address its unique challenges.

## 3 RELATED WORK

**Geospatial Foundation Models (GFM):** We focus on geospatial tasks in this paper, building on a vibrant field of geospatial deep learning that has led to the development of powerful foundation models like Prithvi (Blumenfeld, 2023) and SatMAE (Cong et al., 2023). These utilize large-scale satellite imagery through self-supervised learning to model spatial patterns, and are well-suited to tasks requiring insights from remote sensing data such as landcover segmentation (Robinson et al., 2020). Unfortunately, fine-tuning models in a one-shot setting makes it difficult to adapt to new or unseen regions and times without extensive retraining. More discussions of GFM are in Appendix.

**RL-based Approaches for Active Geospatial Search (AGS)** AGS frameworks, which apply RL to optimize exploration strategies, have been explored in several related geospatial settings (Sarkar et al., 2023a; 2024a;b). These methods typically depend on large-scale labeled datasets to learn effective exploration strategies by simulating search episodes. Additionally, these approaches are often confined to narrow, spatially localized, task-specific settings, such as detecting particular objects (e.g., cars) in satellite imagery. In contrast, our work tackles the challenge of geospatial search with limited data in an open-world context, in which the search is spatially broad, and we must strategically acquire data as well as learn to identify the target of interest - which may be a diffuse phenomenon such as water pollution - via policy learning. Broader discussions on Geospatial Monitoring are in the Appendix.

**Active, Online, Meta, and Lifelong Learning**  Active learning selects informative samples to reduce labeling cost (Ren et al., 2021; Cacciarelli & Kulahci, 2023), and online learning enables continuous updates as new data arrives (Hoi et al., 2021; Shalev-Shwartz, 2011). However, both typically assume stationary task distributions, limiting their effectiveness in dynamic geospatial settings. Lifelong learning (LL) (Kirkpatrick et al., 2017; Shin et al., 2017) addresses non-stationarity via replay, task boundaries, or regularization. Recent advances have begun addressing more constrained settings: methods such as (Jung et al., 2020) avoid replay buffers; task-free and prototype-based methods like those by (Aghasanli et al., 2025) operate without explicit task labels or large buffers. However, these approaches still assume either static selection pools, or environments where sampled examples remain accessible. In contrast, our OWL-GPS setting imposes strict sampling budgets, prohibits revisitation, and requires policy-driven selection, rendering LL methods incompatible without major adaptation.

## 4 METHODOLOGY

**Overview**  To efficiently tackle the OWL-GPS task, we propose a modular framework comprising a concept encoder, a relevance encoder, and a decoder that integrates both for decision-making. The relevance encoder and decoder are trained via an online meta-learning objective, supported by a novel meta-training set formation strategy. Additionally, we introduce a novel active sampling mechanism used during both training and inference, aimed at maximizing sample-efficient learning during training and effective target discovery during inference. We illustrate the high-level overview of our framework in Fig. 2 with a more detailed framework diagram in appendix.

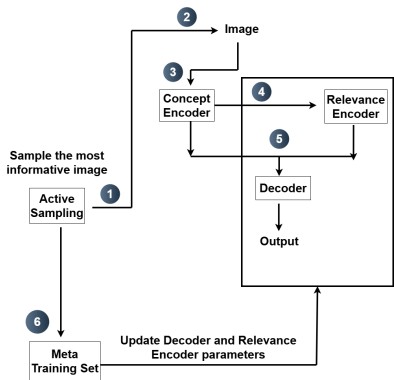

Figure 2: Block Diagram of Framework

**Concept Encoder**  We identify domain-relevant factors influencing target presence, often known from prior studies but variable across regions and tasks. We adopt a generative framework for geospatial search to address data scarcity and distribution shifts, modeling latent region-specific concepts - such as environmental drivers - as transferable priors. By identifying key domain factors from prior studies and encoding each as a latent variable within a generative model (Blumenfeld, 2023; Kingma et al., 2013), we learn low-dimensional concept embeddings that capture underlying geospatial structure from both labeled and unlabeled data. Since these factors are available beforehand, we pre-train an autoencoder-style model to learn latent *concept* representations $\tilde{c}_k(x)$ for each of the $K$ factors characterizing a region $x$, yielding a concept vector $\tilde{c}(x) = [\tilde{c}_1(x), \ldots, \tilde{c}_K(x)]$.

To foster diversity and reduce redundancy, we apply the *Gram–Schmidt orthogonalization* process (GS (Leon et al., 2013)) to produce orthogonal representations $c(x) = GS(\tilde{c}(x))$. These final concept vectors guide sampling and prediction. The implementation details of the concept encoder, empirical results validating the GS transformation, and robustness tests where concepts are randomly masked at inference time are provided in the Appendix.

**Relevance Encoder and Decoder**  It becomes essential to uncover how each concept contributes to shaping the target in a given region $x$. Given that the true *relevance* of each concept is inherently unobservable and may vary across regions and time, it is both intuitive and principled to model *relevance* (denoted as $r(x)$) as a latent variable within a conditional variational autoencoder framework (Kingma et al., 2013; Sohn et al., 2015). Specifically, given a concept vector $c(x)$, we aim to compute the conditional probability of region $x$ containing target, which can be factorized as:

$$\pi_{\theta^0}\big(y \mid c(x)\big) = p\big(y \mid c(x)\big) = \mathbb{E}_{r\big(c(x)\big) \sim p_{\zeta^0}\big(r(c(x))\big)} \left[ p_{\phi^0}\big(y \mid c(x), r(c(x))\big) \right] \qquad (2)$$

Here, the policy parameter $\theta^0$ consists of a relevance encoder parameterized by $\zeta^0$, along with a decoder parameterized by $\phi^0$ that predicts $y$ based on the *relevance* and *concept*. Here, $r\big(c(x)\big) \in \mathbb{R}^K$ denotes the relevance vector and is represented as $r\big(c(x)\big) = [\alpha_{c_1(x)}, \alpha_{c_2(x)}, \ldots, \alpha_{c_K(x)}]$, where $i$'th component, i.e., $\alpha_{c_i(x)}$, quantifies the contribution of the $i$-th concept to the presence of the target in region $x$. We present the implementation details of the relevance encoder and decoder, along with a comparison to simpler alternatives against our CVAE-based formulation, in the Appendix section.

**Proposition 1.** *Optimizing 2 is equivalent to minimizing the following objective:*

$$\min_{\theta=(\phi,\zeta)} \mathbb{E}_{r\left(c(x^{(j)})\right)\sim p_{\zeta^0}\left(r(c(x^{(j)}))\right)} \left[ \log p_{\phi^0}\left(y^{(j)} \mid c(x^{(j)}), r(c(x^{(j)}))\right) \right]$$
$$- \mathrm{KL}\left(p_{\zeta^0}\left(r(c(x^{(j)}))\right) \| p(r(c(x^{(j)})))\right)$$

Derivation of Prop. 1 in the Appendix. Up to this point, we have focused exclusively on optimizing the search policy parameters ($\theta^0$) for inferring the likelihood of a target in a region, conditioned on the concept vector $c(x)$ corresponding to a specific region $x$. However, greedily updating the policy on the latest queried samples can be suboptimal, as it ignores the impact of future observations and may steer the policy in a direction misaligned with upcoming updates. To overcome this, we adopt an online meta-learning approach that equips the search policy with adaptability to future updates while promoting generalization across diverse and evolving geospatial environments.

**Meta-training Set Formation** As opposed to a typical meta-learning setup (Finn et al., 2017), we introduce *core* and *reservoir* buffers to support sample selection. All previously queried labeled samples are stored in a fixed-capacity core buffer. To ensure effective usage before eventual eviction, we assign each sample a score $\frac{duration}{count+1}$, where *duration* is the number of query steps since the sample's insertion, and *count* is the number of times it has been selected for meta-training. A sample is evicted when it exceeds a defined *lifespan* to create space for the newly queried sample at time step $(t+1)$. *Each sample is represented by a compact relevance vector, which we use for greedy clustering in latent space to form structurally diverse meta-batches* (alternate clustering strategy results and greedy clustering strategy details in Appendix). Specifically, the core buffer contributes to forming the meta-training batch by selecting samples from each cluster as follows (See Fig. 3):

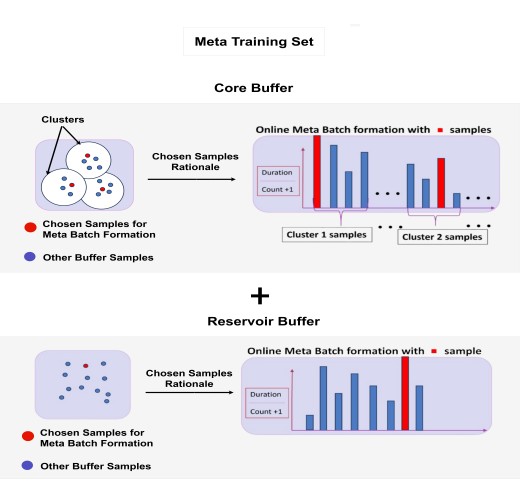

Figure 3: Meta-training Set Formation

$$\mathcal{D}_t^{\text{core}} = \left\{ x_c^* \mid x_c^* = \arg\max_{x\in\mathcal{E}_c}\left(\exp\left(\frac{\text{duration}_x}{\text{count}_x+1}\right)\right), \forall c \in \{1,\dots,N_c\} \right\} \tag{3}$$

Here, $N_c$ denotes the total number of clusters, $\mathcal{E}_c$ represents the set of elements belonging to cluster $c$, and $\mathcal{D}_t^{\text{core}}$ is the set of samples selected from the core buffer for meta-training at time $t$.

When the core buffer reaches capacity, samples exceeding a lifespan threshold are evicted and stored in a reservoir buffer, which provides a secondary opportunity for inclusion. This ensures that samples discarded earlier, though temporarily deemed less relevant, are not permanently excluded, as they may hold significant value under future model parameterizations, supporting long-term learning and adaptation. We sample $K$ elements from the reservoir using the following selection criterion:

$$\mathcal{D}_t^{\text{reservoir}} \sim \text{Sample}_K^{\text{w/o replacement}}\left(\left\{ \frac{\exp\left(\frac{\text{duration}_i}{\text{count}_i+1}\right)}{\sum_{j=1}^N \exp\left(\frac{\text{duration}_j}{\text{count}_j+1}\right)} \right\}_{i=1}^N\right) \tag{4}$$

where $N$ is the total number of reservoir samples. Analogous to the core buffer, the reservoir buffer enforces a lifespan-based eviction policy upon reaching capacity. The final meta-training batch combines both buffers: $\mathcal{D}_t = \mathcal{D}_t^{\text{core}} \cup \mathcal{D}_t^{\text{reservoir}}$, enabling our framework to balance exploration and reuse of informative samples over time.

**Online-Meta Policy Update Rule at t-th Query Step** Our goal is to develop a search policy capable of navigating the challenges posed by the OWL-GPS problem setting with efficiency. To achieve this, it's crucial that the model continuously adapts to data arriving sequentially through query interactions. Embracing this dynamic environment, we adopt an online-meta learning approach (Finn

et al., 2019) that evolves the model parameters on the fly. The meta-training set, curated through our proposed selection strategy, guides this adaptive process. At the $t$-th query step, the optimization objective for training the policy is defined as:

$$\tilde{\pi}_{\theta^{t-1}} = \arg\min_{\pi_{\theta^{t-1}}} \mathcal{L}\left(\pi_{\theta^{t-1}} - \nabla\mathcal{L}(\pi_{\theta^{t-1}}, \mathcal{D}_t^{\text{tr}}), \mathcal{D}_t^{\text{te}}\right)$$

$$\pi_{\theta^t} = \arg\min_{\tilde{\pi}_{\theta^{t-1}}} \tilde{\pi}_{\theta^{t-1}} - \nabla\mathcal{L}\left(\tilde{\pi}_{\theta^{t-1}}, (x^{(q_t)}, y^{(q_t)})\right) \quad (5)$$

Here, $\mathcal{D}_t$ denotes the meta training batch selected following our proposed strategy to update the model parameter at the $t$-th query step. We randomly split $\mathcal{D}_t$ to form the training and evaluation sets, i.e., $\mathcal{D}_t = \mathcal{D}_t^{\text{tr}} \cup \mathcal{D}_t^{\text{te}}$. Here, $\mathcal{L}$ represents pixel-wise binary cross-entropy loss. Following each query, we integrate the most recent observation $(x^{(q_t)}, y^{(q_t)})$ into the core buffer, while evicting a data point following *lifespan* protocol. Next, we present our sampling strategy, which leverages the updated search policy to guide exploration.

**Sampling Strategy During Training** The central objective during training is to sample data that most effectively supports the learning of the search policy, an aim that aligns with the foundational principle of active learning. Consequently, following standard active learning (Cacciarelli & Kulahci, 2023), we select the data point for labeling (denoted as $x^{(q_t)}$) where the model exhibits the highest uncertainty. Our decision-making framework integrates two key modules: a relevance encoder that derives relevance from concepts, and a decoder that localizes potential targets using both relevance and concept. Thus, we sample images by maximizing the combined uncertainty of the relevance and decoder modules, promoting effective co-learning of both components, as illustrated in Fig. 4. This is formalized as follows:

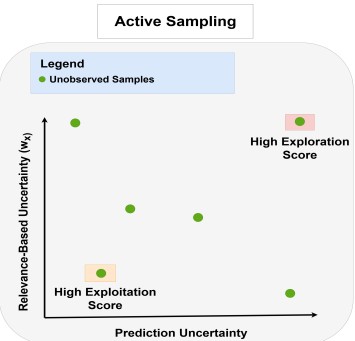

Figure 4: Active Sampling Strategy.

$$x^{(q_t)} = \text{argmax}_{x \in \{X_{train} \setminus \{x^{(q_1)},...,x^{(q_{t-1})}\}\}} \underbrace{\exp^{w_x}}_{\text{relevance uncertainty}} \cdot \sum_{i=1}^{i=P} \underbrace{\left(\exp^{(-|\pi_{\theta^{t-1}}(x_i)-0.5|)}\right)}_{\text{prediction uncertainty}}$$

$$\text{Where, } w_x = \sum_{x\prime \in \{x^{(q_1)},...,x^{(q_{(t-1)})}\}} ||\mu_x - \mu_{x\prime}||_2^2 \quad (6)$$

Here, $\pi_{\theta^{t-1}}(x_i)$ represents the probability that the $i$-th pixel of region $x$ belongs to the target, and $P$ denotes the total number of pixels within region $x$. $\mu_x$ represent the mean of the relevance vector $(r(c(x)))$ computed via the relevance encoder, corresponding to region $x$. As defined in 6, a sample $x$ is prioritized for querying when the model exhibits the highest combined uncertainty of the relevance encoder and the decoder across the pixels within region $x$. We present a Theorem and its proof in the Appendix that lays the ground for the sampling objective during training as defined in 6.

**Sampling Strategy During Inference** During inference, our goal shifts to efficiently discovering target-rich regions under a fixed query budget. To do so, we balance exploitation - focusing on regions with a high predicted likelihood of target presence, and exploration - identifying informative regions.

We first compute the *exploitation score* for an unobserved region $x$ at step $t$ as:

$$\text{Exploit}^{\text{score}}_{\pi_{\theta^{t-1}}}(x) = \exp^{-w_x} \cdot \sum_{i=1}^{i=P} \exp^{\left(\pi_{\theta^{t-1}}(x_i)\right)} \quad (7)$$

This score is high when the updated policy $\pi_{\theta_{t-1}}$ assigns high confidence to multiple pixels in $x$, indicating a strong likelihood of target presence, and when the region exhibits strong similarity to previously queried samples. We present a theorem along with its proof in the Appendix that justifies the rationale behind our exploitation score formulation as defined in 7.

Next, we compute the *exploration score* to encourage querying regions that are dissimilar from previously sampled ones:

$$\text{Explore}^{\text{score}}_{\pi_{\theta^{t-1}}}(x) = \exp^{w_x} \cdot \sum_{i=1}^{i=P} \exp^{(-|\pi_{\theta^{t-1}}(x_i)-0.5|)} \quad (8)$$

This score peaks when predictions are uncertain (near 0.5), and the region differs from past queries, promoting diversity in the sampling process.

To guide selection, we combine these scores using a budget-aware trade-off controlled by $\kappa(C)$:

$$\text{Score}_{\pi_{\theta^{t-1}}}(x) = \kappa(\mathcal{C}) \cdot \text{Explore}^{\text{score}}_{\pi_{\theta^{t-1}}}(x) + (1 - \kappa(\mathcal{C})) \cdot \text{Exploit}^{\text{score}}_{\pi_{\theta^{t-1}}}(x) \tag{9}$$

Here, $\kappa(C)$ is a design choice that decreases over time to emphasize exploration early on and shift toward exploitation as the budget depletes. A simple linear form, $\kappa(\mathcal{C}) = \frac{\mathcal{C}-t}{\mathcal{C}+t}$, is used in practice. After computing the scores (as defined in 9) for each unlabeled image $x$, our approach selects the region with the highest score for sampling.

## 5    EXPERIMENTS

**Evaluation Metrics**    Since OWL-GPS seeks to maximize the identification of locations containing the target of interest, we assess performance using a *Success Rate (SR)* metric as defined below (assuming uniform query cost, i.e., $c(i,j) = 1$):

$$\text{SR} = \sum_{t=1}^{\mathcal{C}} \frac{1}{\min\{\mathcal{C}, U_t\}} \sum_{i=1}^{P} y_i^{(q_t(\pi_{\theta^{t-1}}))} \cdot \mathbf{1}\left[ y_i^{(q_t(\pi_{\theta^{t-1}}))} = \left[ \pi_{\theta^{t-1}}\left( x^{(q_t(\pi_{\theta^{t-1}}))} \right) \right]_i \right]; \tag{10}$$

Here, $U_t$ denotes the maximum number of target pixels in the image queried at step $t$ by the policy $\pi_{\theta_{t-1}}$. The SR metric evaluates how effectively the model selects regions that truly contain the target of interest. Specifically, it computes the proportion of selected regions that are both predicted and *actually* confirmed to be target regions, normalized by the available number of targets in each queried image. This ensures that the metric accounts not just for correct predictions, but also for the *efficiency of query selection* under a constrained sampling budget.

SR is particularly appropriate for the OWL-GPS task, where the overarching objective is to discover as many distinct target regions as possible with limited queries. Unlike standard metrics like accuracy or precision, SR directly reflects the success of the search policy in correctly identifying target-rich regions, which is crucial in real-world deployment scenarios where labeling is expensive and spatial generalization is difficult.

To complement SR, we also report standard classification metrics - *accuracy, F-score, precision*, and *recall* - to assess the model's ability to distinguish target from non-target pixels after querying. We evaluate performance in two settings for the PFAS task: (1) spatial generalization, where training and test data come from different geographic regions within the same year (2019), and (2) temporal generalization, where the model is trained on 2019 data and tested on 2021 data to assess robustness to distributional shift. Dataset details, training setup, compute resources and code details are provided in the Appendix. The Time and memory complexity analysis are also provided in the Appendix.

**Baselines:**    We compare against 6 baselines spanning greedy search, active learning, bandit-based selection, and meta-learning (Table 1). These are selected to reflect diverse modeling assumptions under our OWL-GPS constraints. Detailed discussions on baselines are in Appendix. [1]

---

[1] Traditional active learning methods such as BALD (Houlsby et al., 2011), CoreSet (Sener & Savarese, 2017), or BADGE (Ash et al., 2019) assume access to a static unlabeled pool and the ability to batch-select and retrain. In contrast, our setting involves streaming geospatial inputs, strict sampling budgets, and no revisitation, making such methods infeasible without major adaptation. Our baselines instead span key axes relevant to OWL-GPS: greedy scoring (GA, Prithvi) for single-pass search; UCB for bandit-style adaptive exploration; standard AL adapted to constrained sampling; and both online and active meta-learning to test the value of rapid adaptation. Together, they capture memory-free, task-free, and exploration-aware alternatives within OWL-GPS's constraints.

Table 1: Summary of baseline methods used for comparison.

| Baseline | Description |
|---|---|
| **Greedy Approach (GA)** | Greedy selection using a trained model's confidence. |
| **Prithvi** | Foundation model with one-shot greedy prediction using large-scale geospatial pretraining (Blumenfeld, 2023). |
| **UCB (Auer et al., 2002)** | Bandit-based exploration–exploitation balance using Upper Confidence Bound. |
| **Active Learning (AL)** | Uncertainty-based sampling using a fixed supervised model (Cacciarelli & Kulahci, 2023). |
| **Active Meta-Learning (AML)** | Meta-batch sampling via latent sample representations from prior samples (Kaddour et al., 2020). |
| **Online Meta-Learning (OML)** | Continual updates via single-sample meta-adaptation without episodic batching (Finn et al., 2019). |

Figure 5: *SR* over Different Timesteps

Figure 6: Exploration Behavior for the 2019 PFAS data.

Table 2: Comparison with Baselines (We report Mean $\pm$ Standard Deviation, and are measured over 3 independent trials).

| Year | Method | Acc. | F-score | Precision | Recall | SR |
|---|---|---|---|---|---|---|
| LC | GA | 46% $\pm$ 1.2 | 35% $\pm$ 1.0 | 48% $\pm$ 0.9 | 40% $\pm$ 1.3 | 47% $\pm$ 1.4 |
| | AL | 58% $\pm$ 1.8 | 41% $\pm$ 1.5 | 50% $\pm$ 1.2 | 47% $\pm$ 1.9 | 60% $\pm$ 2.0 |
| | Prithvi | 61% $\pm$ 2.0 | 40% $\pm$ 1.4 | 51% $\pm$ 1.5 | 56% $\pm$ 1.7 | 61% $\pm$ 1.8 |
| | UCB | 67% $\pm$ 2.1 | 48% $\pm$ 1.7 | 53% $\pm$ 1.3 | 59% $\pm$ 1.5 | 68% $\pm$ 2.3 |
| | AML | 53% $\pm$ 2.0 | 40% $\pm$ 1.3 | 65% $\pm$ 1.6 | 30% $\pm$ 1.8 | 54% $\pm$ 2.1 |
| | OML | 67% $\pm$ 1.7 | 49% $\pm$ 1.6 | 82% $\pm$ 1.8 | 39% $\pm$ 1.9 | 67% $\pm$ 1.5 |
| | ***Ours*** | **73% $\pm$ 1.4** | **55% $\pm$ 1.1** | **56% $\pm$ 1.3** | **69% $\pm$ 1.2** | **74% $\pm$ 1.0** |
| 2021 | GA | 58% $\pm$ 1.6 | 44% $\pm$ 1.3 | 45% $\pm$ 1.5 | 44% $\pm$ 1.4 | 68% $\pm$ 2.2 |
| | AL | 61% $\pm$ 1.3 | 39% $\pm$ 1.2 | 45% $\pm$ 1.0 | 33% $\pm$ 1.5 | 66% $\pm$ 1.7 |
| | Prithvi | 64% $\pm$ 1.7 | 48% $\pm$ 1.4 | 48% $\pm$ 1.3 | 48% $\pm$ 1.1 | 75% $\pm$ 1.8 |
| | UCB | 72% $\pm$ 1.5 | 42% $\pm$ 1.2 | 40% $\pm$ 1.1 | 44% $\pm$ 1.3 | 87% $\pm$ 1.9 |
| | AML | 77% $\pm$ 2.0 | 50% $\pm$ 1.5 | 50% $\pm$ 1.7 | 50% $\pm$ 1.4 | 86% $\pm$ 2.1 |
| | OML | 54% $\pm$ 1.9 | 39% $\pm$ 1.4 | 45% $\pm$ 1.6 | 38% $\pm$ 1.7 | 59% $\pm$ 1.9 |
| | ***Ours*** | **80% $\pm$ 0.9** | **53% $\pm$ 1.0** | **58% $\pm$ 1.2** | **53% $\pm$ 1.1** | **95% $\pm$ 0.8** |
| 2019 | GA | 57% $\pm$ 1.5 | 45% $\pm$ 1.3 | 49% $\pm$ 1.4 | 48% $\pm$ 1.7 | 60% $\pm$ 2.0 |
| | AL | 66% $\pm$ 1.8 | 45% $\pm$ 1.5 | 45% $\pm$ 1.3 | 44% $\pm$ 1.6 | 77% $\pm$ 2.1 |
| | Prithvi | 55% $\pm$ 1.6 | 36% $\pm$ 1.3 | 38% $\pm$ 1.4 | 34% $\pm$ 1.5 | 67% $\pm$ 1.8 |
| | UCB | 42% $\pm$ 1.3 | 38% $\pm$ 1.1 | 49% $\pm$ 1.2 | 48% $\pm$ 1.4 | 39% $\pm$ 1.6 |
| | AML | 72% $\pm$ 2.0 | 49% $\pm$ 1.4 | 49% $\pm$ 1.5 | 50% $\pm$ 1.3 | 86% $\pm$ 2.0 |
| | OML | 66% $\pm$ 1.7 | 45% $\pm$ 1.2 | 46% $\pm$ 1.4 | 43% $\pm$ 1.5 | 77% $\pm$ 1.9 |
| | ***Ours*** | **74% $\pm$ 2.0** | **49% $\pm$ 2.0** | **49% $\pm$ 2.0** | **50% $\pm$ 2** | **86% $\pm$ 1.0** |

**Comparison with data from the year 2019**  We first evaluate our method on the 2019 PFAS dataset (Huerta et al., 2018), covering diverse U.S. regions. As shown in Figure 5, our model progressively improves its Success Rate (SR), reflecting an increasing shift toward exploitation. Table 2 confirms that our policy consistently outperforms all baselines.

**Uncovering a Specific Land Cover type within a Strict Sampling Budget**  To test generalization, we apply our framework to land cover (LC) data (Dewitz et al., 2021), targeting the water class despite visually similar categories (e.g., ice). Using a sparsified dataset (details in Appendix), our model maintains SR above 80% under tight supervision (Figure 5), and shows strong predictive accuracy (Table 2), demonstrating adaptability to low-data settings.

**Comparison using data spanning the years 2019 and 2021**  We further evaluate transferability across time by training on 2019 PFAS data and testing on 2021 (U.S. EPA, 2024). Our method significantly outperforms baselines in SR (Fig. 5) and predictive performance (Table 2), validating its effectiveness in spatiotemporally evolving OWL-GPS settings.

**Ablations:**  For the following comparisons, we utilize PFAS data from the year 2019. Additional ablation studies are included in the Appendix to examine the sensitivity of our method to critical hyperparameters, such as buffer sizes, $\kappa(\mathcal{C})$, and varying sampling budgets. We also analyze the effects of core design choices, with further analyses and additional ablations provided in the appendix.

Table 3: Baseline Weaknesses in OWL-GPS

| Baseline | Key Limitations in OWL-GPS Setting |
|---|---|
| UCB | Assumes independent arms; fails under spatial and concept correlations. Leads to inefficient exploration and poor exploitation. |
| AL | Adapts traditional uncertainty sampling to OWL-GPS, but without relevance or feedback-driven correction, often selects uncertain yet uninformative regions. |
| GA | Exploits high-confidence regions early, with no exploration. Gets stuck in local optima. |
| Prithvi | Static, no query refinement or adaptation. Strong pretraining but lacks interactive learning for sequential decision-making. |
| AML | Relies on task identity and few-shot setup; fails in continuous geospatial streams without task boundaries or structure. |
| OML | Lacks relevance filtering; updates on all samples, Leads to noisy adaptation. |

**Effectiveness of Relevance Encoder**
To assess the Relevance Encoder's contribution, we conduct an ablation study in which it is entirely removed from the proposed framework. In this

Table 4: **Importance of Relevance Encoder.**

| Method | Accuracy | F-score | Precision | Recall | SR ($\mathcal{C} = 50$) |
|---|---|---|---|---|---|
| No-RE | 23% ± 1.4 | 23% ± 1.7 | 44% ± 1.5 | 44% ± 1.8 | 13% ± 2.0 |
| **Ours** | **74% ± 1.5** | **49% ± 1.0** | **49% ± 1.7** | **50% ± 2.0** | **86% ± 1.0** |

modified setup, all concepts are assumed to have equal importance. We then compare the performance of this variant (No-RE) with our full proposed framework and report the results in Table 4. We observe that the performance of the No-RE variant is substantially lower than that of our proposed approach, with a performance gap of 73% in terms of SR. Furthermore, the inclusion of the Relevance Encoder consistently enhances the predictability of the search policy (see Table 4). These empirical findings underscore the critical role of the RE in learning an efficient search policy.

**Analyzing the Effectiveness of Meta-training Set Formation**  To evaluate the effectiveness of the proposed meta-training set formation strategy, we compare the search and the predictability performance of our method against a variant in which training is conducted identically, except that samples are randomly selected from a fixed-size core buffer rather than using those chosen through our meta-training strategy. Table 5 presents the SR and the correspond-

Table 5: **Importance of Meta-Training Set.**

| Meta-set | Accuracy | F-score | Precision | Recall | SR ($\mathcal{C} = 50$) |
|---|---|---|---|---|---|
| Random | 49% ± 2.0 | 43% ± 2.1 | 47% ± 1.6 | 47% ± 1.7 | 50% ± 1.4 |
| **Ours** | **74% ± 1.2** | **49% ± 1.5** | **49% ± 0.9** | **50% ± 2.3** | **86% ± 1.0** |

ing predictability metrics. These empirical results underscore that dynamically constructing the meta-training set after each observation - using our proposed formation strategy - leads to substantial gains in both the predictability and search performance (SR) of the learned search policy, reinforcing the utility of our proposed meta-training set formation strategy.

**Probing the Impact of Relevance-Guided Sampling**  To isolate the impact of relevance in shaping exploration and exploitation, we perform an ablation where sampling scores rely solely on the decoder's output, excluding all relevance-informed components.  This stripped-down

Table 6: **Importance of Relevance-Guided Sampling.**

| Meta-set | Accuracy | F-score | Precision | Recall | SR ($\mathcal{C} = 50$) |
|---|---|---|---|---|---|
| No-RG | 60% ± 1.9 | 45% ± 1.7 | 47% ± 2.0 | 46% ± 1.7 | 67% ± 1.5 |
| **Ours** | **74% ± 1.2** | **49% ± 1.1** | **49% ± 1.4** | **50% ± 0.8** | **86% ± 0.7** |

setup reveals the true influence of the relevance space.  Our findings, as reported in 6, show a clear performance drop without relevance-aware guidance (No-RG), highlighting that weighting predictive uncertainty by relevance uncertainty not only enhances sampling efficiency (higher SR) but also ensures improved prediction.

**Analyzing the Exploratory Nature of the Search Policy**  We examine the exploration behavior of the policy across different stages of the active search process. Interestingly, during the initial phase, the policy tends to sample more regions without the target, reflecting its exploratory nature. However, as the search progresses, it increasingly focuses on regions with the target, indicating a shift toward exploitation. These observations, as in Fig. 6, suggest that the learned policy effectively balances exploration and exploitation - an essential requirement for success in OWL-GPS.

## 6   INTERPRETABILITY OF THE PROPOSED FRAMEWORK

**Interpreting Decision Making through the Lens of Concept-Relevance**  We visualized the relevance vectors generated by our CVAE for both correctly and incorrectly predicted samples. We present this visualization in the figure 7. For correctly predicted regions, the top-10 highest weighted concepts consistently correspond to well-established PFAS drivers, such as landfills, airports,

military zones, and land cover, that are strongly supported by environmental literature. In contrast, incorrectly predicted samples have top-ranked concepts less aligned with known PFAS risk factors (e.g., healthcare or financial/insurance areas), offering insight into the reasons for misclassification. This analysis highlights the CVAE's capacity to capture meaningful environmental associations underlying prediction accuracy.

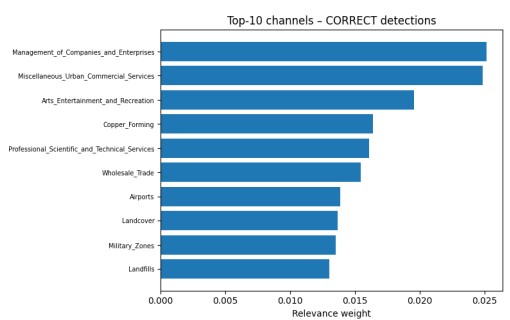 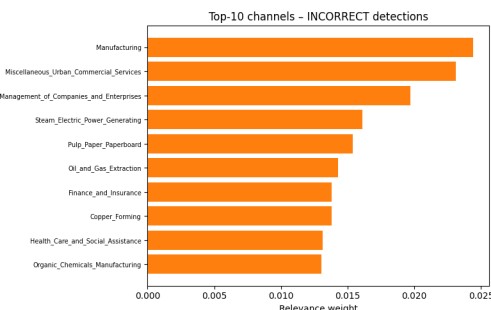

(a) Top-10 Channels of a Correctly Detected Sample    (b) Top-10 Channels of Incorrectly Detected Sample

Figure 7: Interpreting Decision Making through the lens of Concept-Relevance

**Saliency Map Visualization**    We present saliency-trajectory visualizations for Rounds 0, 5, and 10 during inference, as depicted in figure 8, which clearly illustrate the model's transition from scattered early exploration to concentrated exploitation in later rounds. Saliency consistently clusters around the designated target, marked by the central dot, demonstrating robust identification of physiologically relevant regions linked to the known PFAS-positive hotspot.

Beyond local behavior, these plots reveal that the middle-to-bottom portions of the patch become increasingly continuous and spatially coherent over subsequent rounds, corresponding to hydrologically connected waterways near industrial PFAS dischargers, as confirmed by our distance-to-industry heatmap. The model persistently attributes influence to these downstream paths, resulting in contiguous bright regions. While some remaining bright areas may indicate additional true PFAS hotspots, the lack of field samples and environmental cues in those regions motivates the OWL-GPS setting.

These saliency trajectories reveal the model's shift from exploration to exploitation and show how learned relevance patterns can guide targeted field sampling, prioritizing areas that are both predictive and environmentally plausible contamination pathways.

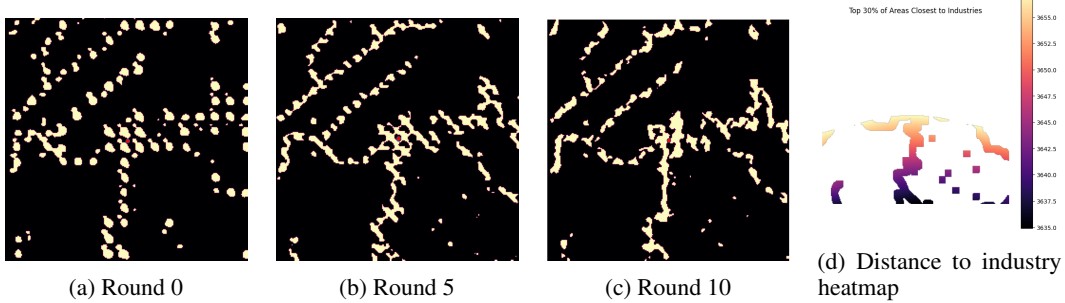

(a) Round 0             (b) Round 5             (c) Round 10             (d) Distance to industry heatmap

Figure 8: Heatmap Evolution across different discovery stages.

Additional interpretability discussions of our proposed framework are provided in the Appendix.

**Conclusion and Limitations**    We introduce a modular, interpretable framework for geospatial target discovery under strict sampling budgets, integrating entropy-based sampling with online meta-learning to adapt to sparse and evolving data. The approach scales well in real PFAS mapping and sparsified land-cover settings, showing strong performance in data-constrained environments. While it relies on domain-specific drivers, which may limit use in unstructured tasks, these drivers are common in many geospatial applications making the framework broadly applicable.

**Reproducibility Statement**    We have taken several steps to ensure the reproducibility of our work. All relevant details about dataset construction, model architecture, training procedures, hyperparameter settings, evaluation metrics, and ablation studies are thoroughly described in the main paper and Appendix. We additionally report the compute resource usage too.

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

# APPENDIX: ADAPTING ACTIVELY ON THE FLY: RELEVANCE-GUIDED ONLINE META-LEARNING WITH LATENT CONCEPTS FOR GEOSPATIAL DISCOVERY

## A  EFFECT OF $\kappa(\mathcal{C})$ ON SEARCH PERFORMANCE

We conduct experiments to assess the impact of $\kappa(\mathcal{C})$ on the search policy's active search performance. This evaluation is carried out using PFAS contamination data from the year 2019 only. Specifically, we investigate how amplifying the exploration weight, by setting $\kappa(\mathcal{C}) = \max\{0, \kappa(\alpha \cdot \mathcal{C})\}$ with $\alpha > 1$, and enhancing the exploitation weight by setting $\alpha < 1$, influence the overall effectiveness of the approach. We present SR results for $\alpha \in \{0.2, 1, 5\}$ as shown in Table 7. The best performance

Table 7: Effect of $\kappa(\mathcal{C})$

| SR Performance across varying $\alpha$ with $\mathcal{C} = 50$ | | |
|---|---|---|
| $\alpha = 0.2$ | $\alpha = 1.0$ | $\alpha = 5.0$ |
| 88% | **95%** | 93% |

is achieved with $\alpha = 1$, and the results suggest that extreme values of $\alpha$ (either too low or too high) hinder performance, as both exploration and exploitation are equally crucial to achieve superior performance.

## B  CONNECTION BETWEEN ACTIVE SAMPLING AND VARIANCE IN RELEVANCE SPACE

In this analysis, we aim to assess the effectiveness of the active sampling strategy. To achieve this, we compute the component-wise variance of the relevance vectors corresponding to the set of samples selected via the proposed active sampling strategy at two different query steps during the active sampling process at inference time. Our observations reveal a progressive decline in batch variance as the search advances, indicating a shift in the model's behavior from exploration to increased exploitation. This behavior is illustrated in Figure 9, 10, which highlights the effectiveness of the proposed sampling strategy. This evaluation is carried out using PFAS contamination data from the year 2019 only.

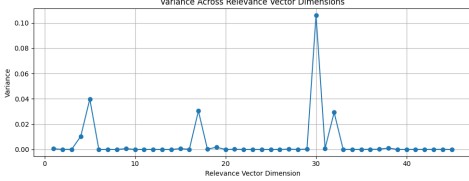

Figure 9: Variance across dimensions of the relevance vector during the initial Active Discovery Phase.

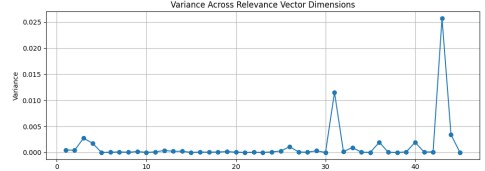

Figure 10: Variance across dimensions of the relevance vector during the later Active Discovery Phase.

## C  EVOLUTION OF DISCRIMINATIVE POWER IN RELEVANCE SPACE AS SEARCH PROGRESS

To assess the effectiveness of our training paradigm, we investigate how the relevance space evolves throughout the active search process. We project the learned relevance vectors into two dimensions using t-SNE at two distinct training stages. Initially, the embeddings are densely entangled, showing little class-specific structure. Note that in our case, we only have two distinct classes, namely class 0 and class 1, where class 0 refers to the absence of the target region, and class 1 indicates the presence of the target region. However, as more informative samples are acquired through active querying, we observe a clear separation in embeddings corresponding to different input classes. This progressive disentanglement highlights the growing discriminative capacity of the relevance space, validating

our approach's ability to refine representations in a data-efficient manner and ensuring increasingly reliable and robust search performance. We present the illustrative visualization in Figure 11, 12 using randomly selected two samples from two distinct classes. This evaluation is carried out using PFAS contamination data from the year 2019 only.

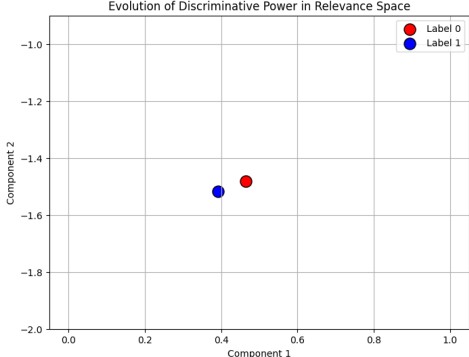

Figure 11: TSNE Visualization of Relevance Vectors during initial Active Discovery Phase.

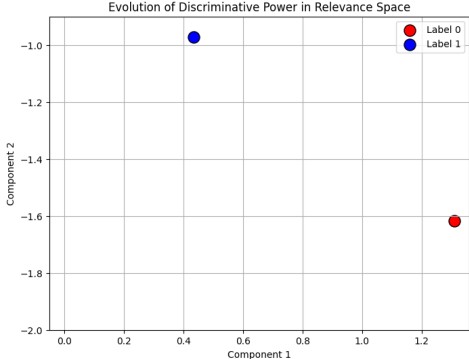

Figure 12: TSNE Visualization of Relevance Vectors during later Active Discovery Phase.

## D    PROOF OF PROPOSITION 1

Optimizing 2 is equivalent to minimizing the following objective:

$$\min_{\theta=(\phi,\zeta)} \mathbb{E}_{r\left(c(x^{(j)})\right) \sim p_{\zeta^0}\left(r(c(x^{(j)}))\right)} \left[\log p_{\phi^0}(y^{(j)} \mid c(x^{(j)}), r(c(x^{(j)})))\right]$$
$$- \mathrm{KL}\left(p_{\zeta^0}\left(r(c(x^{(j)}))\right) \| p(r(c(x^{(j)})))\right)$$

*Proof.* Our primary objective is to maximize $\log p_\theta(y \mid c(x))$, i.e. prediction of targetness from the given set of concepts corresponds to a specific region $x$.

We can marginalize the marginal log-likelihood as follows:

$$\log p(y \mid c(x)) = log \int_{r(c(x))} p(y, r(c(x)) \mid c(x)) \ dr(c(x))$$

By introducing a variational distribution $q_\zeta(r(c(x))$, we can re-write the above expression as:

$$\log p_\theta(y \mid c(x)) = \log \int_{r\left(c(x)\right)} q_\zeta(r(c(x) \mid c(x)) \ \frac{p_\phi(y, r(c(x)) \mid c(x))}{q_\zeta(r(c(x) \mid c(x))} \ dr(c(x))$$

Following the Definition of Expectation, we can write:

$$\log p_\theta(y \mid c(x)) = \log \left[\mathbb{E}_{r(c(x)) \sim q_\zeta((r(c(x)|c(x)))}[\frac{p_\phi(y, r(c(x)) \mid c(x))}{q_\zeta(r(c(x) \mid c(x))}]\right]$$

Now, by applying Jensen's Inequality, we can write it as follows:

$$\log p_\theta(y \mid c(x)) \geq \mathbb{E}_{r(c(x)) \sim q_\zeta((r(c(x)|c(x)))} \left[\log\left[\frac{p_\phi(y, r(c(x)) \mid c(x))}{q_\zeta(r(c(x) \mid c(x))}\right]\right]$$

We can express the above relation as follows:

$$\log p_\theta(y \mid c(x)) \geq \mathbb{E}_{r(c(x)) \sim q_\zeta((r(c(x)|c(x)))} \left[p_\phi(y, r(c(x)) \mid c(x))\right]$$
$$- \mathbb{E}_{r(c(x)) \sim q_\zeta((r(c(x)|c(x)))} \left[q_\zeta(r(c(x) \mid c(x))\right]$$

We can decompose the joint distribution and rewrite it as follows:

$$\log p_\theta(y \mid c(x)) \geq \underbrace{\mathbb{E}_{r(c(x)) \sim q_\zeta((r(c(x)|c(x)))}\left[p(r(c(x)) \mid c(x))\right]}_{1}$$

$$+ \underbrace{\mathbb{E}_{r(c(x)) \sim q_\zeta((r(c(x)|c(x)))}\left[p_\phi(y \mid r(c(x)), c(x))\right]}_{2}$$

$$- \underbrace{\mathbb{E}_{r(c(x)) \sim q_\zeta((r(c(x)|c(x)))}\left[q_\zeta(r(c(x) \mid c(x)))\right]}_{3}$$

By combining term **1** and **3**, we can finally express the objective function as:

$$\log p_\theta(y \mid c(x)) \geq \mathbb{E}_{r(c(x)) \sim q_\zeta((r(c(x)|c(x)))}\left[p_\phi(y \mid r(c(x)), c(x))\right]$$

$$- KL(q_\zeta(r(c(x) \mid c(x)) || p(r(c(x)) \mid c(x)))$$

$\square$

## E    SEARCH PERFORMANCE ACROSS VARYING SEARCH BUDGET

In this section, we present the search performance of our proposed approach under different search budgets, summarized in Table 8. Our results show that increasing the search budget consistently improves performance. As the search budget increases, additional ground truth data becomes available, enabling more effective model parameter optimization. This improved optimization enhances predictive accuracy, which in turn strengthens the overall search performance. This evaluation is carried out using PFAS contamination data from the year 2021.

Table 8: **Search Performance across varying search budget $\mathcal{C}$ with data from the year 2019.**

| Search Budget ($\mathcal{C}$) | Accuracy | F1-score | SR |
|---|---|---|---|
| 30 | 48% | 38% | 52% |
| 40 | 58% | 45% | 62% |
| 50 | **80%** | **53%** | **95%** |

## F    ON THE ROBUSTNESS OF THE CONCEPT ENCODER

To rigorously assess the robustness of the concept encoder, we randomly mask out concept dimensions during inference, effectively simulating real-world scenarios where certain concept information is partially unavailable. We present the empirical findings in the Table 9. Notably, despite these omissions, the approach maintained remarkably consistent accuracy and success rates, with only a marginal drop in overall performance. These findings highlight the strong generalization and practical reliability of the model, demonstrating its capacity to learn resilient and meaningful representations that remain effective even when faced with incomplete or noisy concept inputs - an essential property for robust deployment in diverse and unpredictable environments. This evaluation is carried out using PFAS contamination data from the year 2021.

Table 9: **On the Robustness of the Concept Encoder.**

| Configurations | Accuracy | F1-score | SR |
|---|---|---|---|
| Masking Random Concept during Inference | 75% | 49% | 88% |
| Full Set of Concepts available during Inference | **80%** | **53%** | **95%** |

## G    INFLUENCE OF CORE AND RESERVOIR BUFFER SIZES ON SEARCH PERFORMANCE

In this section, we analyze the impact of the core and reservoir buffer sizes on the search performance. We present the results in Table 10. Our results indicate that performance remains stable across a wide range of buffer sizes, reflecting minimal sensitivity to precise parameter tuning. Furthermore, experiments with varying core–reservoir allocations under a fixed total capacity yielded consistent outcomes, thereby reinforcing the robustness of the proposed approach. This evaluation is carried out using PFAS contamination data from the year 2021.

Table 10: **Search Performance across varying search budget $\mathcal{C}$ with data from the year 2019.**

| Buffer Configurations | Accuracy | F1-score | SR |
|---|---|---|---|
| Core > Reservoir | 83% | 55% | 95% |
| Core = Reservoir | 83% | 55% | 95% |
| Core < Reservoir | 80% | 53% | 95% |

## H    PROOF OF THEOREM 1

**Theorem 1.** *Assuming $k = \{X_{train} \setminus \{x^{(q_1)}, \ldots, x^{(q_{t-1})}\}\}$ represents the set of un-queried datapoints at search step t, and that all previously observed regions have equal importance ($w_i = w_j, \forall i, j$), then the datapoint with the highest entropy $x^{(q_t)}$ can be computed by:*

$$\arg\max_{x^{(q_t)} \in \{X_{train} \setminus \{x^{(q_1)}, \ldots, x^{(q_{t-1})}\}\}} \left[ \log \sum_{i=0}^{(t-1)} \exp\left( \frac{\sum_{x^{(q_t)} \in k} \left(\mu_{x^{(q_t)}} - \mu_{x^{(q_i)}}\right)^2}{2\sigma_{x^{(q_t)}}{}^2} \right) \right],$$

*$\mu_{x^{(q_i)}}$ and $\sigma_{x^{(q_i)}}$ represent the mean and variance of the datapoint queried at i-th step.*

*Proof.* In this section, we present the theoretical justification behind our proposed relevance-guided exploration objective as defined in Equation 6. To start with, we view each point in the relevance space as a sample from a standard Gaussian distribution with mean $\mu_{x^{(q_t)}}$ and variance $\sigma_{x^{(q_t)}}$ corresponding to the sample $x^{(q_t)}$. Without loss of generality, we can express it as follows:

$$p(x^{(q_t)}) = \mathcal{N}(x; x^{(q_t)}, \sigma_{x^{(q_t)}}^2 I)$$

Let the Gaussian Mixture Model with all the observed samples and the unobserved sample $x^{(q_t)}$ can be defined as:

$$p(y) = \sum_{i=1}^{(t-1)} w_i \, \mathcal{N}(y; \mu_{x^{(q_i)}}, \sigma_{x^{(q_i)}}^2 I)$$

where each sample has mean $\mu_{x^{(q_i)}}$ and covariance $\sigma_{x^{(q_i)}}^2 I$. As in our setting, all the previously observed samples are equally important, so we assign $w_i = 1$ for all the samples.

We seek its (differential) entropy. Following the definition of entropy $H$, we can express it as follows:

$$H[p] = -\int p(y) \log p(y) \, dy$$

Plugging it into the Gaussian Mixture Model, we obtain

$$H[p] = -\int \left( \sum_{i=1}^{(t-1)} w_i \, \mathcal{N}(y; \mu_{x^{(q_i)}}, \sigma_{x^{(q_i)}}^2 I) \right) \log \left( \sum_{j=1}^{(t-1)} w_j \, \mathcal{N}(y; \mu_{x^{(q_j)}}, \sigma_{x^{(q_j)}}^2 I) \right) dy$$

Rewriting the order of summation and integration, we obtain:

$$H[p] = - \sum_{i=1}^{(t-1)} w_i \int \mathcal{N}(y; \mu_{x^{(q_i)}}, \sigma^2_{x^{(q_i)}} I) \log \left( \sum_{j=1}^{N_p} w_j \, \mathcal{N}(y; \mu_{x^{(q_j)}}, \sigma^2_{x^{(q_j)}} I) \right) dy$$

Next, we utilize the rule of Gaussian Product Kernel Evaluation.

For two isotropic Gaussians, we can express it as follows:

$$\mathcal{N}(y; \mu_i, \sigma^2 I) \mathcal{N}(y; \mu_j, \sigma^2 I) \propto \mathcal{N}(\mu_i; \mu_j, 2\sigma^2 I)$$

Note that, in the above formulation, we assume that the variances are the same for two different samples in the relevance space. So the kernel integral becomes:

$$\int \mathcal{N}(y; \mu_i, \sigma^2 I) \mathcal{N}(y; \mu_j, \sigma^2 I) \, dy = (2\pi\sigma^2)^{-d/2} \exp \left( -\frac{1}{4\sigma^2} \|\mu_i - \mu_j\|^2 \right)$$

According to the (Hershey-Olsen Bound) (Hershey & Olsen, 2007), approximating the expectation by centering at each mean ($x \leftarrow \mu_{x^{(q_i)}}$), we get:

$$H[p] \approx - \sum_{i=1}^{(t-1)} w_i \log \left( \sum_{j=1}^{(t-1)} w_j \, \mathcal{N}(\mu_{x^{(q_t)}}; \mu_{x^{(q_j)}}, 2\sigma^2_{x^{(q_t)}} I) \right) + \text{const}$$

By expressing the Gaussian with explicit norm:

$$\mathcal{N}(\mu_{x^{(q_t)}}; \mu_{x^{(q_j)}}, 2\sigma^2_{x^{(q_t)}} I) = \frac{1}{(2\pi\sigma^2_{x^{(q_t)}})^{d/2}} \exp \left( -\frac{\|\mu_{x^{(q_t)}} - \mu_{x^{(q_j)}}\|^2}{4\sigma^2_{x^{(q_t)}}} \right)$$

By dropping the normalization constants (absorbed into const), and doubling the denominator to match the common usage:

$$H[p] \approx \text{const} + \sum_{i=1}^{(t-1)} w_i \log \left( \sum_{j=1}^{(t-1)} w_j \, \exp \left( -\frac{\|\mu_{x^{(q_t)}} - \mu_{x^{(q_j)}}\|^2}{2\sigma^2_{x^{(q_t)}}} \right) \right)$$

Hence, the entropy is proportional to:

$$H[p] \propto \sum_{i=1}^{(t-1)} w_i \log \left( \sum_{j=1}^{(t-1)} w_j \, \exp \left( -\frac{\|\mu_{x^{(q_t)}} - \mu_{x^{(q_j)}}\|^2}{2\sigma^2_{x^{(q_t)}}} \right) \right) \tag{11}$$

As we assume each observed sample is equally important, thus, $w_i = w_j = 1$.

Setting $w_i = w_j = 1$, we can write the above expression as:

$$H[p] \propto \log \left( \sum_{j=1}^{(t-1)} \exp \left( -\frac{\|\mu_{x^{(q_t)}} - \mu_{x^{(q_j)}}\|^2}{2\sigma^2_{x^{(q_t)}}} \right) \right)$$

Hence, utilizing the above expression, we can query an unobserved sample $x^{(q_t)}$ that corresponds to maximum entropy. We can choose $x^{(q_t)}$ based on the following criteria:

$$\arg_{x^{(q_t)} \in \{X_{train} \setminus \{x^{(q_1)}, \dots, x^{(q_{t-1})}\}\}} \max \left[ \log \sum_{i=0}^{(t-1)} \exp \left( \frac{\sum_{x^{(q_t)} \in k} (\mu_{x^{(q_t)}} - \mu_{x^{(q_i)}})^2}{2\sigma_{x^{(q_t)}}^2} \right) \right],$$

$\square$

## I  PROOF OF THEOREM 2

> **Theorem 2.** *Assuming policy parameter, $\theta^{t-1}$, the expected log-likelihood of unqueried data $x^{(q_t)}$ containing target $y^{(q_t)}$ can be expressed:*
>
> $$\exp\left\{ -\frac{\sum_{i=0}^{(t-1)}(\mu_{x^{(q_t)}} - \mu_{x^{(q_i)}})^2}{2\sigma_{x^{(q_t)}}^2} \right\} \left( \sum_{i=0}^{P} \exp^{\pi_{\theta^{t-1}}(x_i^{(q_t)})} \right)$$

*Proof.* The expected log-likelihood of unqueried data $x^{(q_t)}$ is composed of two terms. Let's first analyze the first term. We start with the definition of entropy $H$:

$$\mathbb{E}_{x^{(q_t)}}[\log p(x^{(q_t)}|\tilde{x}_{t-1})] = -H(x^{(q_t)}|\tilde{x}_{t-1})$$

Here, $\tilde{x}_{t-1}$ represents the set of already observed datapoints, i.e., $\tilde{x}_{t-1} = \{X_{train} \setminus \{x^{(q_1)}, \ldots, x^{(q_{t-1})}\}\}$. Substituting the expression of $H(x^{(q_t)}|\tilde{x}_{t-1})$ as defined in Equation 11, and by setting $w_i = w_j = 1$, we obtain:

$$\mathbb{E}_{x^{(q_t)}}[\log p(x^{(q_t)}|\tilde{x}_{t-1})] \propto -\log\left( \sum_{j=1}^{(t-1)} \exp\left\{ \frac{||\mu_{x^{(q_t)}} - \mu_{x^{(q_j)}}||_2^2}{2\sigma_{x^{(q_t)}}^2} \right\} \right)$$

$$\mathbb{E}_{x^{(q_t)}}[\log p(x^{(q_t)}|\tilde{x}_{t-1})] \propto -\sum_{j=1}^{(t-1)} \log\left( \exp\left\{ \frac{||\mu_{x^{(q_t)}} - \mu_{x^{(q_j)}}||_2^2}{2\sigma_{x^{(q_t)}}^2} \right\} \right)$$

By simplifying, we obtain,

$$\mathbb{E}_{x^{(q_t)}}[\log p(x^{(q_t)}|\tilde{x}_{t-1})] \propto -\sum_{j=1}^{(t-1)} \left\{ \frac{||\mu_{x^{(q_t)}} - \mu_{x^{(q_j)}}||_2^2}{2\sigma_{x^{(q_t)}}^2} \right\}$$

By leveraging the monotonicity property of the exponential function, we can rewrite the above expression as:

$$\mathbb{E}_{x^{(q_t)}}[\log p(x^{(q_t)}|\tilde{x}_{t-1})] \propto \exp\left\{ -\frac{\sum_{j=1}^{(t-1)}||\mu_{x^{(q_t)}} - \mu_{x^{(q_j)}}||_2^2}{2\sigma_{x^{(q_t)}}^2} \right\} \tag{12}$$

The above expression resembles the first term of the expected log-likelihood as stated in the proposed theorem. It is important to note that our decision-making policy is composed of two parts: (a) the relevance encoder that computes $\mu_{x^{(q_t)}}$ and $\sigma_{x^{(q_t)}}^2$; (b) the relevance decoder that computes the likelihood of containing target $y^{(q_t)}$ given $\mu_{x^{(q_t)}}$ and $\sigma_{x^{(q_t)}}^2$. Which can be represented as follows:

$$\mathbb{E}_{x^{(q_t)}}[\log p(y^{(q_t)}|x^{(q_t)}, \tilde{x}_{t-1})] \propto \sum_{i=0}^{P} \pi_{\theta^{t-1}}(x_i^{(q_t)}) \tag{13}$$

Note, $i$ denotes the pixel index, and $P$ is the number of pixels in $x^{(q_t)}$. Our final objective is to compute the expected log-likelihood of a given geospatial region $x^{(q_t)}$ containing the target $y^{(q_t)}$, which can be expressed as:

Which we can rewrite as follows:

$$\mathbb{E}_{x^{(q_t)}}[\log p(x^{(q_t)}|\tilde{x}_{t-1}) \cdot \log p(y^{(q_t)}|x^{(q_t)}, \tilde{x}_{t-1})]$$

By utilizing the expression of 12 and 13, we can express the above quantity as follows:

$$\underbrace{\mathbb{E}_{x^{(q_t)}}[\log p(x^{(q_t)}|\tilde{x}_{t-1}) \cdot \log p(y^{(q_t)}|x^{(q_t)}, \tilde{x}_{t-1})]}_{\textit{Expected Log-likelihood of } x^{(q_t)} \textit{ containing the target } y^{(q_t)}} \propto \exp\left\{-\frac{\sum_{j=1}^{(t-1)} ||\mu_{x^{(q_t)}} - \mu_{x^{(q_j)}}||_2^2}{2\sigma^2_{x^{(q_t)}}}\right\}$$

$$\cdot \sum_{i=0}^{P} \pi_{\theta^{t-1}}(x_i^{(q_t)})$$

By utilizing the monotonicity property of the exponential function, we can rewrite the above expression as:

$$\underbrace{\mathbb{E}_{x^{(q_t)}}[\log p(x^{(q_t)}|\tilde{x}_{t-1}) \cdot \log p(y^{(q_t)}|x^{(q_t)}, \tilde{x}_{t-1})]}_{\textit{Expected Log-likelihood of } x^{(q_t)} \textit{ containing the target } y^{(q_t)}} \propto \exp\left\{-\frac{\sum_{j=1}^{(t-1)} ||\mu_{x^{(q_t)}} - \mu_{x^{(q_j)}}||_2^2}{2\sigma^2_{x^{(q_t)}}}\right\}$$

$$\cdot \sum_{i=0}^{P} \exp^{\pi_{\theta^{t-1}}(x_i^{(q_t)})}$$

$\square$

## J   DETAILS OF GREEDY INTERSECTION CLUSTERING ALGORITHM

This section details the greedy-intersection clustering algorithm used to cluster the relevance vectors in the core buffer, as highlighted in the meta-training set formation subsection in the main paper. The Greedy Intersection clustering algorithm was originally proposed in Ge et al. (2025). We present it in detail for completeness. Consider a set $\theta \in T$ of relevance vectors, fix $K$ (i.e., number of clusters), and suppose $\epsilon$ is the radius of each cluster. The key intuition behind the Greedy Intersection Algorithm is that for any $\theta \in T$, an $\epsilon$-hypercube centered at $\theta$ characterizes all possible $\theta'$ that can cover $\theta$ in the sense that $||\theta - \theta'||_\infty \leq \epsilon$. Thus, if any pair of $\epsilon$-hypercubes centered at $\theta$ and $\theta'$ intersects, any point at the intersection covers both. To illustrate, consider the following simple example:

$$[x_3, x_4] \qquad\qquad\qquad x_4 - \epsilon \;\vdash\!\!\!\!\times\!\!\!\!-\!\!-\; x_4 + \epsilon$$
$$[x_1, x_2, x_3] \qquad\qquad x_3 - \epsilon \;\vdash\!\!-\!\!\times\!\!-\!\vdash\; x_3 + \epsilon$$
$$[x_1, x_2] \qquad x_2 - \epsilon \;-\!\!-\!\!\times\!\!-\; x_2 + \epsilon$$
$$[x_1] \qquad x_1 - \epsilon \;-\!\!-\!\!\times\!\!-\!\vdash\; x_1 + \epsilon$$

Each cross marks a target parameter to be covered, and each line segment shows the range within which an $\epsilon$-close representative of that parameter may lie. Selecting a point within the overlapping region of these intervals allows us to simultaneously cover multiple parameters.

The Greedy Intersection algorithm builds on this intuition by generalizing it as follows. In the first stage, it constructs an intersection tree independently for each dimension. For the $s$-th dimension, the data points are sorted in ascending order based on their $s$-th coordinate, denoted as $x_1 < x_2 \cdots < x_n$. For each point $x_i$, we initialize a list containing only $[x_i]$, which will be used to track how many other points can be jointly covered along with it.

Starting from the second smallest datapoint $x_2$, we check if $x_2 - \epsilon \leq x_1 + \epsilon$, i.e. if $x_2 \leq x_1 + 2\epsilon$. Since $x_2 - \epsilon > x_1 - \epsilon$ due to our sorting, any point inside $[x_2 - \epsilon, x_1 + \epsilon]$ can cover both $x_1, x_2$. Therefore if this interval is valid, we add $x_1$ to the list $[x_2]$ to indicate the existence of a simultaneous coverage for $x_1, x_2$. In general, for $x_i$, we check if $x_i \leq x_j + 2\epsilon$ with a descending $j = i - 1$ to 1 or until the condition no longer holds. If the inequality is satisfied, we add $x_j$ to $x_i$'s list. Then since we have ordered the set, for every index $j'$ less than $j$, $x_i > x_j + 2\epsilon > x_{j'} + 2\epsilon$. The coverage for all the $x$ in $x_i$'s list would be the interval $[x_i - \epsilon, x_j + \epsilon]$, where $j$ is the smallest index in $x_i$'s list. There are $1 + 2 + \cdots + n - 1 = \mathcal{O}(n^2)$ comparisons in total. We form a set of these lists, and

call it $\mathcal{A}_s$ for the $s$-th dimension. The figure above illustrates how the algorithm works to find out $\mathcal{A}_1 = \{[x_1], [x_1, x_2], [x_1, x_2, x_3], [x_3, x_4]\}$.

The second stage of the algorithm aims to identify a hypercube that covers the maximum number of points, selecting one axis-aligned interval from each dimension. By the geometry of Euclidean space, two points $\theta_1$ and $\theta_2$ are within an $\epsilon$-distance in $\ell_\infty$ norm if and only if they appear together in each other's lists across all dimensions. Thus, to find a hypercube whose center is within $\ell_\infty$-distance of the most data points, we search for the combination of lists $l_1, \ldots, l_d$, where each $l_s$ is chosen from the corresponding set $\mathcal{A}_s$, such that their intersection has the maximum cardinality. In our illustrative example, we observe that the groups $[x_1, x_2, x_3]$ and $[x_3, x_4]$ must be covered separately, each by a different point positioned between the red or blue vertical lines.

A key limitation of the Greedy Intersection Algorithm (GIA) is its exponential complexity with respect to the number of dimensions in the data. To address the challenge of high dimensionality, we apply standard dimensionality reduction techniques—specifically, Principal Component Analysis (PCA) - to project each data point into a lower-dimensional space that is computationally more tractable. We then apply the Greedy Intersection Algorithm to these PCA-transformed points. In our experiments, we use a reduced dimensionality of 7, which we found to yield strong performance in our problem setting. The complete algorithm is presented below:

---

**Algorithm 1** Greedy Intersection

---

**Input**: $T = \{\theta_i\}_{i=1}^N, \epsilon > 0, K \geq 1$
**Output**: Parameter cover $C$

1:   $C \leftarrow []$
2:   **for** $round\ k = 1$ to $K$ **do**
3:      **for** $dimension\ m = 1$ to $d$ **do**
4:         Sort $T$ in ascending order based on their $m$-th coordinates
5:         $lists_m \leftarrow []$
6:         **for** $indiviual\ i = 2$ to $N$ **do**
7:             $S_i \leftarrow [\theta_i]$
8:             **for** $j = i - 1$ to $1$ **do**
9:                 **if** $\theta_i$'s $m$-th coordinate $< \theta_j$'s $m$-th coordinate $+2\epsilon$ **then**
10:                   Add $\theta_j$ to $S_i$
11:                 **else**
12:                   **if** $lists_m[-1] \subseteq S_i$ **then**
13:                      $lists_m[-1] \leftarrow S_i$
14:                   **else**
15:                      Add $S_i$ to $lists_m$
16:                   **end if**
17:                   **break**
18:                 **end if**
19:             **end for**
20:         **end for**
21:      **end for**
22:      $S^{1*}, \ldots, S^{m*} \leftarrow \text{argmax}_{S^1 \in lists_1, \ldots, S^m \in lists_m} |S^1 \cap \cdots \cap S^m|$
23:      $covered \leftarrow S^{1*} \cap \cdots \cap S^{m*}$
24:      $\hat{\theta}_k \leftarrow$ average of the $covered$
25:      $T \leftarrow T - covered$
26:      $C.\text{adds}(\hat{\theta}_k)$
27: **end for**
28: **return** $C$

---

Additionally, to evaluate the effectiveness of the Greedy Intersection clustering algorithm, we conduct a comparative analysis by replacing it with a standard clustering method (i.e., DBScan) and measuring the resulting performance on the downstream task. This evaluation is carried out using PFAS contamination data from the year 2019 only. The outcomes of this comparison are summarized in the following table. Our empirical results indicate that the framework incorporating the Greedy Intersection clustering approach consistently yields improvements across various evaluation metrics,

albeit slightly, highlighting the validity of the Greedy Intersection clustering approach under the proposed framework.

Table 11: **Effectiveness of the Clustering Approach.**

| Clustering Approach | Accuracy | F-score | SR ($\mathcal{C} = 50$) |
|---|---|---|---|
| DBScan | 68% | 46% | 80% |
| Ours | **74%** | **49%** | **86%** |

# K    ADDITIONAL ANALYSIS ON THE IMPORTANCE OF RELEVANCE GUIDED SAMPLING

Our adoption of a Conditional Variational Autoencoder (CVAE) as the backbone of our relevance estimator is driven by the inherent dynamism of open-world geospatial search environments, where the relevance of concepts can shift unpredictably across both spatial and temporal dimensions. The CVAE framework enables the model not only to learn complex relevance patterns but also to quantify associated uncertainties - empowering the system to make decisions that are both adaptive and robust to changing conditions. This uncertainty-aware modeling is especially crucial in large-scale and data-sparse geospatial scenarios, where assumptions of static or simple relevance easily break down.

In rigorous empirical comparisons, the CVAE-based estimator consistently outperforms a simpler multi-layer perceptron (MLP) attention-based relevance estimation baseline (denoted as Ours (RG $\rightarrow$ MLP)), delivering significant improvements in performance, as demonstrated in the results in Table 12 and also detailed in Section 5 (Tables 4 and 6). These findings underscore the substantial practical advantage of leveraging probabilistic deep generative modeling for relevance estimation, affirming that the CVAE's ability to jointly represent relevance values and their uncertainties translates directly to superior real-world search effectiveness. This evaluation is carried out using PFAS contamination data from the year 2021.

Table 12: **Analyzing the impact of Relevance Guided Sampling.**

| Method | Accuracy | F1-score | SR |
|---|---|---|---|
| Ours (RG $\rightarrow$ MLP) | 68% | 50% | 80% |
| Ours | **80%** | **53%** | **95%** |

# L    IMPORTANCE OF ORTHOGONALIZATION IN THE CONCEPT SPACE

We assess the effectiveness of the orthogonalization layer by comparing the performance of the proposed framework with a variant that only excludes orthogonalization. The results are presented in Table 13. This evaluation is carried out using PFAS contamination data from the year 2019 only. The empirical results indicate a noticeable decline in the metrics when the orthogonalization layer is omitted from the proposed framework, highlighting the importance of the orthogonalization of the concepts within the proposed framework.

Table 13: **Importance of Orthogonalization of Concepts.**

| Orthogonalization | Accuracy | F-score | SR ($\mathcal{C} = 50$) |
|---|---|---|---|
| No | 36% | 35% | 30% |
| Yes (our framework) | **74%** | **49%** | **86%** |

# M    ADDITIONAL ANALYSIS ON THE ORTHOGONALIZATION LAYER

To further evaluate the role of orthogonalization, we compared two variants: (1) using only the concept encoder, and (2) using the concept encoder with Gram-Schmidt (GS) orthogonalization. All other components of the framework were excluded in both cases. Results are shown in Table 14. Our

findings reveal that GS orthogonalization by itself provides only a marginal improvement over the standalone concept encoder. This clearly illustrates that orthogonalization alone is insufficient to drive significant gains. Instead, GS orthogonalization proves effective only when integrated synergistically with the full framework, highlighting the necessity of a comprehensive approach to realize its full potential.

Table 14: **Analyzing the impact of the concept orthogonalization layer.**

| Configuration | Accuracy | F1-score | SR |
|---|---|---|---|
| Only Concept Encoder | 32% | 31% | 26% |
| Only Concept Encoder with Orthogonalization | **35%** | **34%** | **27%** |

## N    ADDITIONAL RESULTS WITH STATE-OF-THE-ART GEOSPATIAL FOUNDATION MODEL

In the main paper, we benchmarked our proposed framework against the state-of-the-art geospatial foundation model, Prithvi-V1. Our empirical results reveal that relying solely on a foundation model is inadequate for addressing the unique challenges of the OWAGS task. To further substantiate this conclusion, we extended our evaluation to include the enhanced Prithvi-V2 model, comparing its performance directly with our approach (see Table 15). Our experiments clearly demonstrate that improving the foundation model alone is insufficient for solving OWAGS efficiently. These findings underscore the substantial advantages and impact of our proposed framework, establishing it as essential for tackling open-world geospatial search challenges with true robustness and effectiveness. This evaluation is carried out using PFAS contamination data from the year 2021.

Table 15: **Comparisons with State-of-the-art Geospatial Foundation Model.**

| Method | Accuracy | F1-score | SR |
|---|---|---|---|
| Prithvi-V1 | 64% | 48% | 75% |
| Prithvi-V2 | 58% | 52% | 53% |
| Ours | **80%** | **53%** | **95%** |

## O    DATASET DETAILS

We evaluate our approach on two geospatial prediction tasks: PFAS contamination prediction and land cover classification, both of which use satellite data aligned with sparsely sampled ground truth.

For the PFAS contamination prediction task, each sample consists of a multi-channel raster patch centered on a known PFAS measurement site from EPA datasets, including the National Rivers and Streams Assessment (NRSA) [2] and the National Lakes Assessment Fish Tissue Study [3]. This dataset is inherently sparse, with only 704 labeled sample points across the US. Each patch is of size 256 × 256 pixels at 30-meter resolution and includes a total of 45 channels. These channels encode diverse geospatial features, such as land cover rasters from the National Land Cover Database (NLCD) [4], flow direction rasters capturing hydrological connectivity obtained from the ArcGis Pro software, and distance transforms to known PFAS discharger sites obtained from the U.S EPA database [5]. This rich set of environmental and spatial layers provides the model with meaningful context to predict contamination.

For the land cover classification task, we use imagery from the Sentinel-2 surface reflectance dataset [6]. For each sample region, we extract 4-channel image patches using the Near-Infrared (NIR), Red,

---

[2]NRSA

[3]National Lakes Assessment Fish Tissue Study

[4]NLCD

[5]US EPA

[6]Sentinel-2

Green, and Blue bands. Each patch is of size 256 × 256 pixels at 30-meter resolution. These patches are aligned with the same spatial footprint as the PFAS patches to ensure consistency in coverage.

## P  DETAILS OF PSEUDOLABEL GENERATION PROCEDURE TO INDUCE STABILITY DURING TRAINING

Given the scarcity of ground truth data in PFAS monitoring, typically limited to sparse point-based measurements, we employ a point-based label expansion strategy to generate dense supervision signals suitable for segmentation training. Specifically, each labeled point is used to generate a pseudo-label mask over its surrounding patch, enabling the model to learn from broader spatial context [7].

For each georeferenced point with a PFAS contamination label (1 for above health advisory threshold, 0 for below), we extract a raster patch centered on that point. All surface water pixels within this patch are assigned the same label as the central point, reflecting the assumption, guided by environmental science, that contamination may be correlated in hydrologically connected regions. Non-surface water pixels are assigned a special label (2) and are excluded from loss computation. This strategy effectively converts sparse point annotations into dense training masks, allowing the model to be trained as a segmentation network despite the absence of full pixel-wise labels.

Crucially, model evaluation is performed exclusively on ground truth point data from the U.S. EPA PFAS datasets, ensuring that performance metrics reflect real-world observations and are not influenced by the assumptions underlying pseudo-label generation.

For the land cover dataset, dense per-pixel land cover annotations are used as supervision only during training, and are derived from public land cover products that include standard classes such as water, forest, urban, and agricultural land [8]. During testing, however, we adopt a sparse label evaluation setup: only one or two labeled pixels are retained in each test mask, simulating the limited label availability often encountered in practice. The model is expected to make predictions over the entire patch, but evaluation is conducted only at these sparsely labeled locations. This setup mirrors the PFAS task and encourages the model to generalize beyond isolated supervision.

### P.1  ADDITIONAL DISCUSSIONS ON THE LABEL COST

The OWL-GPS setting models constraints on supervision, not imagery. For instance, **Sentinel imagery is abundant; however, the labels required for the discovery tasks in our datasets are not. In real scientific monitoring problems like PFAS contamination, labeled examples are scarce, expensive, and slow to acquire.** Moreover, OWL-GPS assumes an online, memory-limited regime in which the agent cannot store or replay past inputs beyond a small fixed buffer. These are the core constraints the framework is designed to capture.

1. PFAS (2019 & 2021): Each PFAS label corresponds to an actual field measurement collected by environmental agencies. Such measurements require field deployment, sample collection, laboratory chemical analysis, and administrative coordination. As a result, very few labeled samples exist, and expanding coverage is constrained by budget, personnel, and lab throughput. This is the source of "costly" and "resource-constrained." In addition, under OWL-GPS assumptions, inputs are non-replayable: once a geospatial region has been processed and leaves the limited memory buffer, it cannot be revisited for training or queried again. The agent therefore must decide which regions to label as they arrive.

2. Land Cover (LC): For LC, labels are not inherently difficult to acquire, but the purpose of including LC is to demonstrate that the framework generalizes beyond PFAS. To place LC into a discovery setting comparable to PFAS, we **sparsify the supervision as described in the Appendix P**. This creates a regime where only a small subset of tiles are labeled, matching the limited-supervision assumption of OWL-GPS. We clarify that this sparsification is simulated to evaluate generalization, not a claim about real-world LC label difficulty.

---

[7]FOCUS

[8]NLCD

## Q    DETAILS OF TRAINING AND INFERENCE HYPERPARAMETERS

We used a patch size of 256×256 pixels for all geospatial regions. Model training was performed using the AdamW optimizer, with learning rate dynamically adjusted using a polynomial decay scheduler with warmup. Training used focal loss with a KL-divergence regularization term as part of a conditional variational autoencoder objective for relevance learning. Meta-updates were performed using meta-batches of size 10, formed by selecting one representative sample per cluster from the core buffer, and three samples from the reservoir buffer. All reported results are averaged over three independent runs with different random seeds to ensure robustness.

## R    ARCHITECTURE DETAILS

### R.1    DETAILS OF CONCEPT ENCODER

This section details the implementation of the concept encoder as highlighted in the concept encoder subsection in the main paper. The Concept Encoder is implemented using a modified Vision Transformer (ViT) architecture, tailored to handle geospatial imagery possibly with temporal structure. The encoder is structured as a deep transformer-based module that ingests high-dimensional multi-spectral or multi-temporal satellite imagery and maps it into a low-dimensional concept space that captures the essential semantics of the input.

The encoder uses a patch-based embedding mechanism to divide the input tensor into non-overlapping spatial patches. Each patch is linearly projected into an embedding vector, to which a learnable class token and fixed sinusoidal 3D positional encodings are added. These embeddings are then processed through a deep stack of transformer blocks, each consisting of multi-head self-attention and MLP layers, followed by a final normalization step.

The encoder is initialized using a sin-cos positional embedding and can optionally load pretrained weights. To improve representation disentanglement and mitigate redundancy, we apply Gram-Schmidt orthogonalization to the learned latent vectors, which promotes interpretability and diversity among the learned concept axes. The output of the encoder is a tuple of patch-wise embeddings enriched with contextual information, which serve as the base representations for relevance inference and downstream target prediction.

### R.2    OVERVIEW OF FRAMEWORK

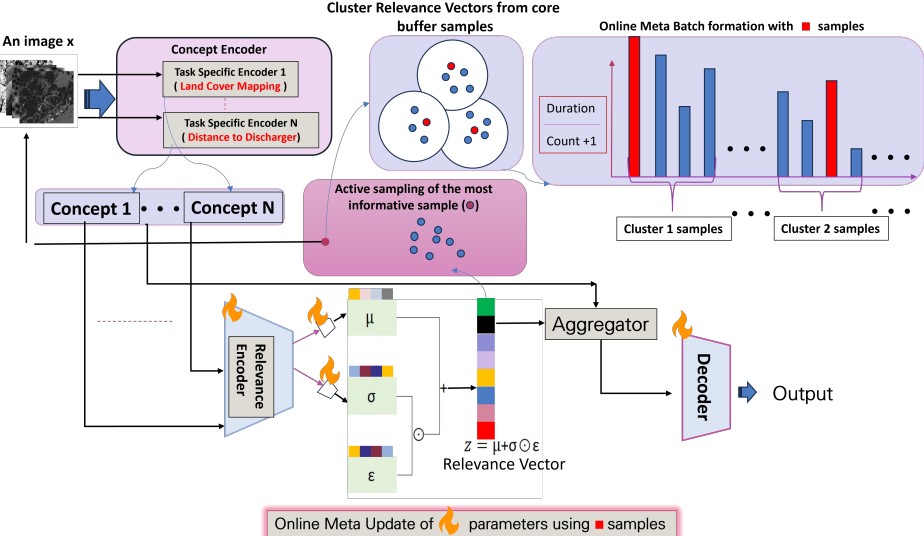

Figure 13: Overview of our proposed framework.

### R.3 DETAILS OF RELEVANCE ENCODER

This section details the implementation of the relevance encoder as highlighted in the relevance encoder and decoder subsection in the main paper. The Relevance Encoder estimates a latent relevance distribution over concepts to capture their varying importance in influencing target presence across geospatial regions. It is modeled as the probabilistic encoder component in a Conditional Variational Autoencoder (CVAE), and takes as input the region-specific concept embeddings from the concept encoder.

This module is designed to process batches of concept embeddings organized per concept, and leverages a lightweight yet expressive convolutional backbone consisting of three convolutional layers with ReLU activations and batch normalization, followed by spatial downsampling via max-pooling. The resulting feature maps are flattened and passed through two separate linear layers to produce the mean and standard deviation vectors, one for each concept. These vectors parameterize a diagonal Gaussian distribution from which a sample relevance vector is drawn during training via the reparameterization trick.

The output relevance vector quantifies the relative contribution and uncertainty of each concept with respect to the target prediction task. This facilitates not only accurate modeling but also reasoning about the influence of different domain factors. Empirically, the relevance encoder was shown to significantly boost predictive accuracy and search efficiency.

## S DETAILS OF THE DECODER

This section details the implementation of the decoder as highlighted in the relevance encoder and decoder subsection in the main paper. The Decoder is responsible for producing pixel-level predictions of target presence by conditioning on the fused representation of concepts and their corresponding relevance scores. It is designed using a dual-head FCN (Fully Convolutional Network) architecture to enhance segmentation robustness during learning.

Each head is a configurable FCN variant that processes the fused feature map via a sequence of convolutional layers, followed by feature concatenation and a final classification layer. The first head is lightweight with a single convolution, while the second includes a deeper convolutional stack to model more complex interactions.

The decoder architecture aligns with semantic segmentation principles and supports variable numbers of output classes, dropout for regularization, and batch normalization to stabilize training. This component plays a critical role in translating abstract latent representations into actionable geospatial predictions.

## T ADDITIONAL INTERPRETABILITY ANALYSIS OF THE PROPOSED FRAMEWORK

Here, we visualized two distinct correctly predicted samples, each highlighting meaningful PFAS-related factors, though their top-ranked concepts vary. We present this visualization in figure 14. This demonstrates that the model dynamically adjusts relevance weighting based on local geographic context, a fundamental principle of the OWL-GPS framework, rather than relying on a uniform global importance pattern. Collectively, these visualizations provide clear evidence that (i) correct predictions associate with semantically meaningful, domain-consistent concepts, (ii) incorrect predictions tend to emphasize less relevant or spurious factors, and (iii) relevance weighting adapts contextually across different geospatial environments.

## U ADDITIONAL DETAILS ABOUT THE BASELINES AND EVALUATION METRIC

### U.1 DETAILS ABOUT THE BASELINES:

OWL-GPS introduces a new constrained, streaming, non-replayable problem formulation. Therefore, each baseline must be carefully adapted to comply with the OWL-GPS constraints, such as streaming

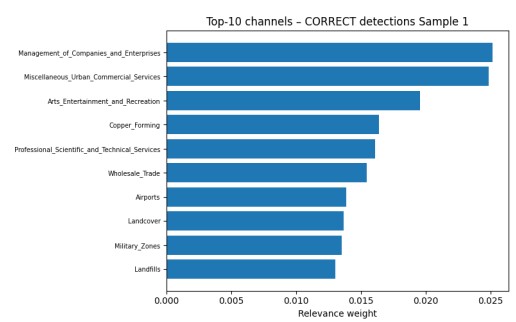 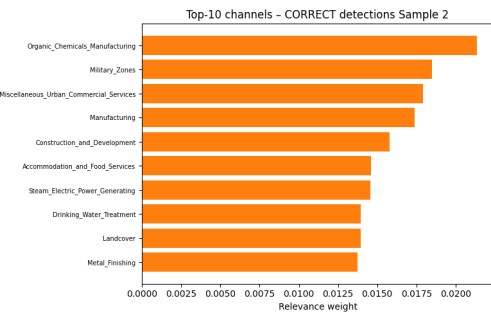

(a) Top-10 Contributing concepts in Region 1.   (b) Top-10 Contributing concepts in Region 2.

Figure 14: Region-aware Decision Making and Analyzing the Mis-detection

inputs, no revisitation, limited memory and query budget, and updates only through queried labels. Below, we provide explicit descriptions of how every baseline is trained and evaluated under this unified setup.

1. **Active Learning (AL)**: Successful performance in OWL-GPS requires a delicate balance between exploration and exploitation during inference. AL methods primarily focus on exploration by querying areas of highest model uncertainty. While this ensures thorough coverage, it overlooks exploiting accumulated knowledge about promising regions. Consequently, AL struggles to efficiently navigate the trade-off necessary for high performance in OWL-GPS. For training with the active learning baseline, we adhere strictly to the standard maximum uncertainty strategy for selecting samples to query during a standard supervised training with a ViT. For evaluation, we apply our sampling strategy with one key modification: the selection of queried samples is driven exclusively by the exploration score, with no consideration given to the exploitation score in the sampling process.

2. **Greedy Approach (GA)**: GA represents the contrasting extreme by prioritizing exploitation — querying locations where the model predicts the highest likelihood of the target. Purely exploitation-driven strategies like GA risk premature convergence to suboptimal regions and fail to sufficiently explore the environment. Our experimental findings corroborate this limitation, showing inferior performance of GA relative to our method. GA conducts a single forward pass over all incoming regions during evaluation after a standard supervised training phase with a ViT, assigning confidence scores to each. It then selects the top-K regions based on the query budget. GA does not perform online updates, store past regions, or revisit inputs, making it a strict single-pass, non-adaptive baseline within OWL-GPS.

3. **Prithvi**: Although Prithvi is a powerful and general framework, it is not specifically designed to handle the multi-faceted challenges of the OWL-GPS problem, such as continual adaptation in dynamic environments, and robust online updating. These complex constraints underscore the need for a tailored approach like ours. Specifically, the geospatial foundation model Prithvi is trained using a standard supervised approach. During evaluation, it assigns confidence scores to unobserved samples, selects the highest scoring sample, updates its parameters with the latest observations, and repeats this process for subsequent batches until the query budget is exhausted.

4. **UCB (Bandit exploration)**: At first glance, the OWL-GPS problem may appear similar to a Multi-Armed Bandit (MAB) scenario, making UCB a natural baseline. However, unlike typical MAB settings where each arm is independent, geospatial environments exhibit strong spatial correlations; observing one location informs us about neighboring regions. UCB does not leverage this spatial structure, treating each arm independently. This lack of spatial modeling fundamentally limits its performance in our problem setting, as reflected in Table 2. We would like to emphasize that UCB is adapted to the OWL-GPS setting by treating each incoming region as an "arm" whose score is computed from its predicted reward (model confidence) and an exploration bonus. Once a region is queried based on the score, the model updates online using the new labeled sample, following the same online meta-update

procedure as in our framework. UCB does not access any historical unlabeled pool, and all decisions are made per-arrival as in our framework.

5. **Online Meta-Learning (OML)**: OML in our setup follows the same online meta-update rule used in our framework. The key difference is that OML does not use our meta-batch formation strategy with GIA for constructing diverse update sets. Instead, OML directly applies the update using the examples present in our buffer at that moment. Aside from the absence of GIA, OML operates identically to our framework under OWL-GPS constraints: streaming inputs, no revisiting beyond the small buffer, and continuous online meta-adaptation from the sparse labeled data encountered so far.

6. **Active Meta-Learning (AML)**: AML follows a training approach similar to ours but differs in two key ways: (a) it forms the meta-training batch using standard random sampling from the buffer, and (b) it lacks an online update mechanism, unlike our proposed strategy. Our experiments demonstrate that AML is ineffective in the OWL-GPS setting, highlighting the critical role of an online update mechanism combined with a coverage-based meta-batch formation strategy.

## V    EXTENDED RELATED WORK

### V.1    ENVIRONMENT MONITORING:

Prior studies such as Sarkar et al. (2023b) and Sarkar et al. (2023a) have investigated active sampling in related environmental monitoring contexts. However, several fundamental constraints, including the need for strictly online operation, adherence to tight budget limits, memory constraints, and the handling of non-stationary and evolving distributions, distinguish our setting and make these previous approaches unsuitable for OWL-GPS. More generally, there is a rich literature on environmental monitoring methods, but the vast majority focus on settings with static or replayable datasets, abundant labeling, and stationary environments, without addressing the unique challenges posed by our real-world, online discovery scenario. Below, we briefly review key environmental monitoring works to further contextualize the distinctiveness of our approach. Environmental monitoring relies heavily on geospatial analysis, an interdisciplinary approach combining geography, computer science, statistics, and engineering to extract valuable insights from spatial data such as satellite imagery, GPS, and historic datasets. This enables critical applications including understanding land use patterns (Park et al. (2023)), infrastructure planning (Amaral et al. (2021); Regona et al. (2024)), and assessing environmental impacts (Milà et al. (2023)). These geospatial tools support urban planning for sustainable expansion (Gharaibeh et al. (2020)) and transportation systems (Kamruzzaman et al. (2015)). Recent technological advances, including improved satellite resolution, enhanced GPS accuracy, and increased drone deployment, have transformed data collection and analysis capabilities, facilitating real-time monitoring of key phenomena such as deforestation (Monjardin-Armenta et al. (2020)), glacier retreat (Thapliyal et al. (2023)), and urban heat islands. Additionally, geospatial analysis supports disaster risk management, including wildfire (Shafapourtehrany et al. (2023)), flood (Liao et al. (2023)), and earthquake prediction (Lam et al. (2021)), improving evacuation and relief efforts (Manfré et al. (2012)).

Machine learning (ML), deep learning (DL), computer vision (CV), and natural language processing (NLP) have been increasingly integrated with geospatial tools to enhance data processing and predictive power (Casali et al. (2022)). These methods facilitate large-scale pattern recognition and predictive modeling, crucial for environmental risk assessment and urban sustainability (Son et al. (2023)). CV enables automated interpretation of satellite and street-view imagery for land classification and change detection (Marasinghe et al. (2024)), whereas NLP analyses research articles and reports for insights on environmental changes and biodiversity trends (Cao et al. (2024)). Overall, this integration of geospatial sciences with advanced computational methods paves the way for innovative environmental monitoring solutions, addressing challenges from urbanization to climate change with greater precision and scale. However, our current OWL-GPS problem setting is fundamentally distinct from previously studied problems, capturing the complex nuances of real-world environmental monitoring challenges in a unified framework.

## V.2 GEO-SPATIAL FOUNDATIONAL MODEL

The development of specialized foundation models has been driven by the need for precision and contextual sensitivity within scientific domains. In geospatial analysis, Geospatial Foundation Models (GFMs) have emerged as powerful tools tailored to interpreting complex Earth surface and atmospheric patterns (Mai et al. (2023)). GFMs address critical challenges such as spatial heterogeneity (Sun et al. (2022)), temporal dynamics (Yao et al. (2023)), and the multidimensional nature of geospatial data (Jakubik et al. (2023)). Transformer architectures, especially Vision Transformers (ViT; Dosovitskiy (2020)) and hierarchical designs like Swin Transformers (Liu et al. (2021)), have become foundational components due to their ability to model long-range dependencies and dynamic attention. Recent innovations enhance GFMs capabilities for spatiotemporal data, for instance, incorporating temporal information as channels (Jakubik et al. (2023)) or designing multi-branch networks to capture spatial affinity and temporal continuity (Yao et al. (2023)). Adaptations of pretrained models such as SAM (Kirillov et al. (2023)) exemplify how conventional image analysis methods are refined for specific geospatial tasks like SAR imagery segmentation (Yan et al. (2023)). Furthermore, Masked Autoencoders (MAE; He et al. (2022)) are widely employed for self-supervised training, enabling scalable learning from unlabeled geospatial imagery, while supervised fine-tuning is applied for specialized downstream tasks requiring category-specific outputs (Yan et al. (2023); Yao et al. (2023)). Among Geospatial Foundation Models, IBM's Prithvi (Jakubik et al. (2023)) is distinguished by its innovative approach to GeoAI and geospatial data analysis, making it a prime candidate for detailed evaluation in this work. Prithvi supports six spectral bands;Red, Green, Blue, NIR, SWIR 1, and SWIR 2, enabling richer spectral information capture beyond conventional RGB imagery. This multi-band capability enhances model versatility and effectiveness across diverse geospatial applications, including land cover mapping and environmental monitoring. Furthermore, Prithvi's extensive pretraining on NASA's Harmonized Landsat and Sentinel dataset empowers it to harness temporal and spatial patterns effectively .(Jakubik et al. (2023)).

## W TIME AND MEMORY COMPLEXITY DETAILS OF THE PROPOSED FRAMEWORK:

### W.1 TIME COMPLEXITY DETAILS:

The time complexity of our proposed approach primarily depends on the meta-training set formation and policy update steps performed at each query:

- At each query step $t$, the meta-training set is dynamically curated from a core buffer and a reservoir buffer, involving clustering and selection based on sample relevance and diversity. Assuming a buffer size of $K$ and relevance embedding dimension $D$, the clustering complexity is approximately $O(K^2 \cdot D)$.
- The meta-policy update consists of gradient-based optimization over the meta-training batch. The complexity scales linearly with the batch size $B$, the parameter count of the concept encoder, relevance encoder, and decoder, as well as the cost of forward and backward passes. Denoting the total cost of forward and backward passes as $P$, this step has complexity $O(B \cdot P)$.
- Each query also requires computing exploitation and exploration scores over the unlabeled pool of size $N$ to select the next sample. The complexity for scoring each candidate region involves $O(N^2 \cdot D + P)$, leading to a total complexity of $O(N^2 \cdot D + N \cdot P)$.

In summary, the overall time complexity per query step can be expressed as:

$$O(K^2 \cdot D) + O(B \cdot P) + O(N^2 \cdot D + N \cdot P) \approx O(N^2 \cdot D) \quad (\text{assuming } K \ll N) \approx O(N^2)$$

(assuming $D \ll N$).

Furthermore, we present the time and memory costs associated with our proposed framework as follows:

- On an NVIDIA A100 GPU, sampling step during inference takes on average approximately 2.1 minutes (128.8 seconds). The model's decision-making step is lightweight, requiring only 4 GB of GPU memory.

## W.2 DETAILS ABOUT COMPUTE RESOURCE AND CODE

All experiments were conducted using PyTorch on an NVIDIA A100 GPU. We also provide the code for reference in the following anonymous GitHub link [9].

# X    LLM USAGE

We used ChatGPT to polish some sentences in the paper.

---

[9]https://anonymous.4open.science/r/OWL-GPS-Code-0D2D

