# OpenReview forum: "Adapting Actively on the Fly: Relevance-Guided Online Meta-Learning with Latent Concepts for Geospatial Discovery"
_ICLR.cc/2026/Conference — ICLR 2026 Conference Desk Rejected Submission_

### Official Review · Reviewer_A8mn · 2025-10-17

**Soundness:** 3
**Presentation:** 2
**Contribution:** 2
**Rating:** 4
**Confidence:** 2

**Summary:**

The paper introduces an online active learning framework for geospatial discovery, combining concept-driven representations, uncertainty-based sampling, and online meta-learning to adapt under strict sampling budgets. The proposed system targets open-world settings where new data continuously arrive and cannot be revisited, addressing challenges of non-stationarity and limited supervision.

**Strengths:**

The paper tackles a timely and challenging problem of online adaptation for geospatial tasks, proposing a unified framework that integrates concept encoding and meta-learning.

**Weaknesses:**

- Regarding interpretability, the study would benefit from visualizing which concepts or POI dimensions are most influential, along with strategy heatmaps or trajectory plots explaining the model’s sampling behavior. Qualitative analysis of failure cases could clarify which factors or uncertainty scores led to misdetections.
- Regarding experiments, significance testing is missing; an ablation that removes the Gram–Schmidt orthogonalization step is needed to validate the contribution of concept diversity and de-correlation.
- Regarding complexity, theoretical and empirical analysis of time and memory costs is lacking.
- Regarding some minor issues:
    - line 222: “Derivation of 1 in the Appendix.” should reference Proposition 1 or Eq. (1).
    - Terminology consistency: unify “Open-World” vs. “Open World.”
    - line 213: `[αc1, αc1, …, αcK]` should be `[αc1, αc2, …, αcK]`.

**Questions:**

- Given that active learning relies on accurate uncertainty estimation, the framework employs a CVAE to model latent relevance r(x) but does not exploit its probabilistic properties. Why explicitly incorporate the posterior variance of r(x) into the sampling utility function? Has any uncertainty calibration been performed?
- Considering that the proposed task involves sequential decision-making, adaptation, and long-term reward maximization, could the active sampling strategy be theoretically formulated within a Meta-RL framework?

---

> ### Author Response · Authors · 2025-11-20
> **Response to Reviewer A8mn**
>
> We thank the reviewer for the constructive evaluation and for recognizing the relevance of the OWL-GPS setting and the contribution of integrating concept encoding with online meta-learning for geospatial adaptation. We appreciate these encouraging remarks and address the reviewer’s comments and suggestions in detail below.
>
> > Q1: Regarding interpretability, the study would benefit from visualizing which concepts or POI dimensions are most influential, along with strategy heatmaps or trajectory plots explaining the model’s sampling behavior. Qualitative analysis of failure cases could clarify which factors or uncertainty scores led to misdetections.
>
> **A1:** To address the reviewer’s request for interpretability, we visualized the relevance vectors produced by our CVAE for both correctly and incorrectly predicted samples **(See Figure1.png in the following link: https://anonymous.4open.science/r/OWL-GPS_VIS-F28E/).** **The top-10 highest weighted concepts for correctly predicted regions consistently include well-established PFAS-related drivers—such as landfills, airports, military zones, and land cover—all of which are strongly supported by environmental literature as major PFAS sources**. In contrast, **incorrectly predicted samples show top-ranked concepts that are relatively less aligned with known PFAS risk factors (e.g., healthcare or financial/insurance–related areas), which helps explain why those regions were misclassified**.
>
> We also visualized two different correctly predicted samples, and while both highlight meaningful PFAS-related factors, the specific top-ranked concepts differ across samples **(See Figure2.png in the following link: https://anonymous.4open.science/r/OWL-GPS_VIS-F28E/)**. This illustrates that **the model adapts its relevance weighting to the local geographic context—a core motivation of the OWL-GPS framework—rather than relying on a single global importance pattern**. Together, these visualizations provide direct evidence that (i) correct detections are associated with semantically meaningful and domain-consistent concept dimensions, (ii) incorrect detections tend to emphasize less informative or spurious drivers, and (iii) relevance weighting is context-adaptive, varying naturally across different geospatial environments.
>
>
> In response to the request for trajectory plots, we include saliency-trajectory visualizations (Rounds 0, 5, and 10 in the inference phase) **(See Figure3.png in the following link: https://anonymous.4open.science/r/OWL-GPS_VIS-F28E/)**. These plots reveal a clear transition from early scattered exploration (Round 0) to **increasingly concentrated exploitation** in later rounds. Notably, the saliency consistently clusters around the target (known PFAS-positive location indicated by the central dot), demonstrating that the model reliably identifies and reinforces regions physiologically linked to the confirmed hotspot.
>
> Beyond this local behavior, we also observe that the **middle-to-bottom portion of the patch becomes more continuous and spatially coherent over rounds** **(See Figure4.png in the following link: https://anonymous.4open.science/r/OWL-GPS_VIS-F28E/)**. This structure aligns with hydrologically connected waterways that are **closest to known industrial PFAS dischargers**, as confirmed by our distance-to-industry heatmap. The model repeatedly attributes influence to these connected downstream paths, creating continuous bright regions. Some of the remaining bright areas may similarly correspond to true PFAS hotspots; however, we lack field samples and other environmental cues in those regions to verify this—precisely motivating our OWL-GPS setting. These trajectory plots therefore illustrate not only the model’s shift from exploration to exploitation but also how the learned saliency patterns can **inform targeted field sampling**, prioritizing regions that are both influential for predictions and environmentally plausible contamination pathways. **We have included these additional interpretability analysis in Section 6  and Appendix T of the revised manuscript.**
>
> > Q2: Regarding experiments, significance testing is missing; an ablation that removes the Gram–Schmidt orthogonalization step is needed to validate the contribution of concept diversity and de-correlation.
>
> **A2** **The Gram-Schmidt (GS) orthogonalization is a critical component of our proposed framework, thoroughly analyzed in Appendix sections L and M**. While GS orthogonalization alone is insufficient to yield significant gains, empirical evidence (see Tables 13 and 14) demonstrates its indispensable role: omitting this step results in substantial degradation in active discovery performance. Our empirical findings also highlight that GS orthogonalization's effectiveness emerges only when integrated synergistically within the full framework.

---

> ### Author Response · Authors · 2025-11-20
> **Response to Reviewer A8mn**
>
> > Q3: Regarding complexity, theoretical and empirical analysis of time and memory costs is lacking.
>
> **A3:** The time complexity of our proposed approach primarily depends on the meta-training set formation and the policy update steps executed at each query:
> * At each query step t, the meta-training set is dynamically curated from a core buffer and a reservoir buffer, involving clustering operations and selection criteria based on sample relevance and diversity. The complexity of this meta-batch formation grows roughly with the size of the buffers and the clustering algorithm used. **Assuming the buffer size of K, dimension of relevance embedding to be D, clustering complexity is O($K^{2}$ $\cdot$ D).**
>
> * The meta-policy update involves gradient-based optimization over the meta-training batch, whose  complexity scales linearly with the batch size (B), the parameter count of the concept encoder, relevance encoder and decoder, and the cost of forward and backward passes. **Assuming the total cost of forward and backward passes of our model to be P, the complexity amounts to O($B$ $\cdot$ $P$).**
>
> * Each query also involves computing exploitation and exploration scores over the unlabeled pool to select the next sample, thus complexity scales linearly with the pool size. **Moreover for each candidate region, computation of exploration/exploitation score requires complexity of O($N^{2}$ $\cdot$ D + P). Hence, the net complexity amounts to  O($N^{2}$ $\cdot$ D + N $\cdot$ P).**
>
> In summary, the overall time complexity per query step is approximately:
> **O($K^{2}$ $\cdot$ D) + O($B$ $\cdot$ $P$) +  O($N^{2}$ $\cdot$ D + N $\cdot$ P) $\approx$ O($N^{2}$ $\cdot$ D)  [assuming K << N] $\approx$ **O($N^{2}$)** [Assuming D << N].**
>
> Furthermore, we present the time and memory costs associated with our proposed framework as follows:
>
> - **On an NVIDIA A100 GPU, sampling step during inference takes on average ≈2.1 minutes (128.8 seconds). The model’s decision-making step is lightweight, requiring only ~4 GB of GPU memory**.
>
> **We have included these detailed discussion in the revised draft (see Appendix section W)**.
>
> > Q4: Regarding minor editorial fix.
>
> **A4:** Thank you for pointing these out. **We have fixed these editorial issues in the updated draft.**
>
> > Q5.a: Given that active learning relies on accurate uncertainty estimation, the framework employs a CVAE to model latent relevance r(x) but does not exploit its probabilistic properties.
>
> **A5a** The framework fully leverages the probabilistic nature of the CVAE's latent relevance r(x), where **the decoder's prediction uncertainty, crucial for active learning, is conditioned on $r(x)=$ $\mu_x$  + $\epsilon$ $\cdot$ $\sigma_x$, sampled from a multivariate Gaussian with mean $\mu_x$ and variance $\sigma_x$.** This mechanism, detailed in **Figure 11 (Appendix)**, enables the framework to exploit the CVAE’s probabilistic properties. Additionally, the posterior variance of r(x) informs the relevance uncertainty, which is integral to the exploration score driving active learning and sample selection.

---

> > ### Author Response · Authors · 2025-11-20
> > **Response to Reviewer A8mn**
> >
> > > Q5.b: Why explicitly incorporate the posterior variance of r(x) into the sampling utility function?
> >
> >  **A5.b:** The explicit incorporation of the posterior variance of r(x) into our sampling utility function is grounded in both theoretical and practical motivations.
> >  * Fundamentally, **the posterior variance of r(x) serves as a direct quantification of epistemic uncertainty in the relevance estimation** provided by our CVAE. While the decoder’s predictive uncertainty captures ambiguity in the outcome given relevance, the latent relevance uncertainty itself encodes how much the model “knows” about the underlying domain-driven factors at each candidate location. **By considering both sources of uncertainty—the predictive entropy from the decoder and the posterior variance from the relevance encoder—we ensure that sampling decisions account for ambiguity in both the mapping from data to relevance and the mapping from relevance to final decision-making.** This dual treatment is crucial in active learning for high-impact scientific discovery, where a single-source uncertainty estimate can often be misleading if the latent variable distribution itself is poorly estimated. In other words, the posterior variance of r(x) is not just a proxy for entropy; it reflects the confidence of the model in its relevance encoding at a given location, directly informing the informativeness of sampling candidates in the context of concept-relevance-guided discovery.
> > * Empirically, **ablation studies reported in Section 5 (see Table 6) robustly demonstrate that removing relevance-aware uncertainty from the acquisition function leads to statistically significant drops in discovery efficiency and sample utility.** By explicitly leveraging this probabilistic property, our framework systematically prioritizes observations that maximize overall model learning, leading to superior performance under constrained budgets and non-stationary distributions.
> >
> > > Q5.c: Has any uncertainty calibration been performed?
> >
> > **A5c** **To ensure robust generalization and open-world deployability, our framework avoids hand-crafted calibration techniques such as temperature scaling or Bayesian recalibration**; instead, the probabilistic nature of the CVAE offers intrinsic, interpretable uncertainty quantification for both relevance and decision outputs. To further illuminate this built-in calibration, we monitor the concentration of learned relevance distributions during active target discovery (**see Appendix B for details**). Our results show a progressive decline in relevance uncertainty variance as search proceeds, signaling a natural transition from exploration to exploitation. This shift, visualized in Figures 7 and 8 (in the Appendix), underscores the effectiveness of our sampling strategy and highlights the framework's inherent uncertainty calibration.
> >
> > > Q6: Considering that the proposed task involves sequential decision-making, adaptation, and long-term reward maximization, could the active sampling strategy be theoretically formulated within a Meta-RL framework?
> >
> > **A6:**   At a high level, one might conceptualize OWL-GPS within a Meta-Reinforcement Learning (Meta-RL) framework, given its sequential decision-making, adaptation, and long-term reward maximization aspects. However, there are significant challenges that impede this formulation. Below we list some of the key reasons behind this:
> > * **Very Limited Training Samples and Sparse Reward:** As outlined in the introduction, the OWL-GPS setting deals with tasks where obtaining ground truth observations is prohibitively costly, limiting us to only about 100 samples—at least two to three orders of magnitude fewer than typical Meta-RL frameworks require. This challenge is compounded by an extremely sparse reward signal, with only very few samples containing the target. Such extreme sparsity renders standard Meta-RL approaches unsuitable for this setting.
> > * **Infeasibility of simulating diverse training trajectories limits efficient RL training.** With only ~100 training samples and an episode length or query budget of ~50, generating diverse, sufficiently long training episodes is impractical, constraining RL effectiveness in this setting.
> > *  **No Task Boundaries:** Standard Meta-RL framework assumes pre-defined task boundaries. However, OWL-GPS task operates under continuously (online) changing geospatial distribution without any pre-specifed task boundaries, limiting the use of Meta-RL framework.

---

### Official Review · Reviewer_wCqr · 2025-10-28

**Soundness:** 3
**Presentation:** 3
**Contribution:** 3
**Rating:** 6
**Confidence:** 3

**Summary:**

Summary: This paper considers a challenge in geospatial machine learning for target indification -- there is often a sparsity of labels to learn from, so on-thy-fly learning is important for algorithmic approaches to adapt to new spatial or temporal domains. At the same time, the number of samples that can be collected is often small, necessitating an explore-exploit tradeoff for continual learning. To model these challenges, the authors present a novel problem formulation, Open-World Learning for Geospatial Prediction and Sampling (OWL-GPS), as well as a meta-learning based approach to solving this problem. They focus on applications in rare-class landcover detection, and PFAS hotspot identification. Overall, the scope of this paper is quite ambitious, contributing to both strengths and weaknesses that I outline below.

NOTE: because many formal statements/proofs are in the appendix I did not have time to check them -- content in the appendix did not factor into my review or score.

**Strengths:**

1. To the best of my knowledge, the problem formulation is novel. Also to the best of my knowledge, the problem formulation captures real-world challenges for online data collection with limited budgets, seen in many environmental monitoring settings.
2. This paper has several contributions: a novel problem formulation, a novel method to address the problem, and a new benchmark for this problem.
3. The proposed method performs better on average than the baselines (none of which are designed for this problem since it's a new problem formulation), some ablation studies are run to test the importance of the main components of the method (meta-training set and relevance-guided sampling). Overall, this gives evidence that the problem formulation is a challenging one needing new methods. It also shows that the proposed approach is working on this problem setting.

**Weaknesses:**

1. There are so many design choices in the proposed methods, but not enough space to explain them, and not enough explain experiments to undertsand what matters. There are ablations on the relecance encoder and meta-training set, but only for one dataset (2019) - why not all three datasets? Does the GS orthogonalization matter? Forgive me if I missed it but I don't think that's tested.
2. Related work is sparse: the part on geospatial foundation models only cites 2 papers? There are a ton. Either do a thorough related work here or specify what scope you're looking at for this part of the related work. Honeslty I'm not sure this part of your related work needs to be there, but maybe I'm missing something. For the active geopsatial search, is Sarkar et al. really the only team that has tackled this? Maybe the other work out there is less directly realted to what this paper tackles, but the readers need to understand what types of similar problems have been studied before in the geo/environemntal montiroing space, and this is too sparse to get that across.
3. No standard deviations or standard errors on any of the results makes it impossible to intepret the statistical significance of the performance benefits.

Formatting/figures/typos

1. What is "a)" in the block diagram figure? This should be clear from the figure and/or legend alone.
2. The font in figure 3 is way too small.
3. Line 333: I think you mean K(C) *decreases* over time
4. equation 10 seems unfinished.
5. Table 4 appears before table 3. Fix the reference numbers so they appear in order
6. The buffer around inset table and figure captions is too small. it's clear the authors are aggressively using \vspace{-something} and it inhibits readability. There are alternatives, for example float table 5 and 6 together at the top or bottom of the page to span both columns.

**Questions:**

1. I have some uncertainty as to what is going on in equation 1 with the notation that uses both q and i to index sampled regions. I think answering the following would help: Why is the problem formuluation done at a pixel level? I might be missing something, but I would think that people annotating satellite images can easily annotate major swatches of an image at a time. At the same time, does the objective in eq. (1) penalize asking for a large swatch of say, n xm  pixels to be labeled, if the target of interest is in that swatch but is small compared to the n x m dimension? If so, that seems a bit odd to me, because you've still found the target of interest. Would this then bias the algorithm toward big targets and away form small ones?
2. The conclusion states that this method is interpreatable. What aspects of the method aid interpretability?
3. could you please correct the formula in eq. 10 and kindly let me know what it should be?

---

> ### Author Response · Authors · 2025-11-20
> **Response to Reviewer wCqr**
>
> We thank the reviewer for the thoughtful and detailed assessment, and we appreciate the recognition of the novelty of the OWL-GPS problem formulation, the methodological contributions, and the overall strength of our experimental evidence. We are grateful for the reviewer’s engagement and address all noted concerns and suggestions point-by-point below.
>
>
> > **Q1** Many design choices in the proposed methods... not enough experiments to undertsand what matters.
>
> **A1** Due to space limitations in the main manuscript, **we have included extensive analysis and comprehensive ablation studies in the supplementary material comprising every design choice we make (Appendix A,B,C,E,F,G,J,K,L,M,N). This supplementary content thoroughly dissects every key design choice and component of our proposed framework.** By providing these detailed experiments and evaluations, we ensure readers can fully understand the impact and necessity of each element, thereby reinforcing the rigor and robustness of our approach.
>
> > **Q2** Does the GS orthogonalization matter?
>
> **A2** Indeed, GS orthogonalization is a critial component in our proposed framework. **We analysed the role of GS orthogonalization in the Appendix section L and M**. Our analysis indicates that GS orthogonalization alone is insufficient to drive significant gains. Instead, GS orthogonalization proves effective only when integrated synergistically with the full framework.
>
> > **Q3** Ablations on the relevance and meta-training set, but only for one dataset (2019) - why not all 3 datasets?
>
> **A3** Core ablations of the relevance encoder, meta-training set, and uncertainty components are presented using the 2019 PFAS dataset, which best represents the real-world complexity and distributional challenges OWL-GPS is designed to address. To complement this, the appendix includes additional ablations with the 2021 dataset, focusing on different framework aspects such as concept perturbation under distributional shift. **Since the PFAS datasets share comparable structure, duplicating all ablations across both would offer limited new insight; instead, our approach enables focused analysis in both stable and shifted contexts. The land cover dataset was used to assess framework generalizability to new modalities and task structures.**
>
> > **Q4** Related work is sparse.
> >
> **A4** The previous related work section focused on the most directly relevant research, but we agree that a broader overview would better contextualize our contribution. Following your suggestion, **in the revised version (Appendix V), we expanded this section to include a more comprehensive set of references covering major geospatial foundation models and related efforts in environmental and geospatial monitoring.**
>
>
>
>
> > **Q5** K\(C) decreases over time?
>
> **A5** Yes. We have fixed this typo in the updated paper.
>
> > **Q6** Comments on formatting/figures/typos
>
> **A6** We've fixed all editorial issues in the revision.
>
> > **Q7** Why is the problem formuluation done at a pixel level?
>
> **A7** The pixel-level formulation in Eq. (1) directly reflects how environmental supervision is obtained in OWL-GPS settings. **Unlike manual image annotation, labels are derived from physical field measurements taken at precise geographic sampling points, which are then mapped to individual pixels in satellite imagery. It is not feasible to obtain ground-truth for an entire region without physically sampling each location; thus, supervision is inherently point-based. Modeling at pixel resolution faithfully represents this data collection process, ensures that only sampled locations inform the model**.
>
> > **Q8** Does the objective in eq. (1) penalize discovering small targets inside a larger swatch.
>
> **A8** The objective in Eq. 1 does not penalize the discovery of small targets within larger regions. When a region is queried, the model receives ground-truth labels for all pixels inside that tile, regardless of target size. Thus, even if the target is small compared to the tile, it is fully supervised and counted as a successful detection. The formulation simply aggregates pixel-level supervision from environmental measurements and does not down-weight tiles containing small targets or favor large contiguous structures. This ensures consistent evaluation without bias against small-scale discoveries.
>
> > **Q9** correct the formula in eq. 10?
>
> **A9** Thank you for noting this. The missing closing outer square bracket in Eq. 10 has been corrected in the revision.
>
>  > **Q10** What is "a)" in the block diagram figure?
>
> **A10** "a)" in the block diagram refers to the components in our framework that are updated, that is, the segmentation decoder head and the relevance encoder. We have updated this figure in the revised version.

---

> ### Author Response · Authors · 2025-11-20
> **Response to Reviewer wCqr**
>
> > **Q11** What aspects of the method aid interpretability?
>
> **A11** Interpretability in our framework arises from the model’s ability to attribute each sampling decision to a transparent, quantitative combination of human-understandable concepts, each assigned a learned relevance score. This design enables domain experts to inspect and trace which concepts most influenced each decision—correct or incorrect—thereby supporting post-hoc analysis and building trust in the system. **Additional interpretability analysis is provided in our response to Reviewer A8mn (we refer to Section 6 and Appendix T of the revised draft for details).**
>
> > **Q12** No standard deviations on any of the results?
>
> **A12** All reported results are averaged over three independent random seeds. We apologize for not mentioning this earlier in the paper. Below, we provide the updated main results table with the corresponding variance bars. We have included this in the revised draft.
>
>
> ### Table 2: Comparison with Baselines (Mean ± Std over 3 seeds)
>
> | Year | Method  | Acc.           | F-score        | Precision       | Recall         | SR             |
> |------|----------|----------------|----------------|------------------|----------------|----------------|
> | LC   | GA       | 46% ± 1.2      | 35% ± 1.0      | 48% ± 0.9        | 40% ± 1.3      | 47% ± 1.4      |
> |      | AL       | 58% ± 1.8      | 41% ± 1.5      | 50% ± 1.2        | 47% ± 1.9      | 60% ± 2.0      |
> |      | AML      | 53% ± 2.0      | 40% ± 1.3      | 65% ± 1.6        | 30% ± 1.8      | 54% ± 2.1      |
> |      | OML      | 67% ± 1.7      | 49% ± 1.6      | 82% ± 1.8        | 39% ± 1.9      | 67% ± 1.5      |
> |      | Prithvi  | 61% ± 2.0      | 40% ± 1.4      | 51% ± 1.5        | 56% ± 1.7      | 61% ± 1.8      |
> |      | UCB      | 67% ± 2.1      | 48% ± 1.7      | 53% ± 1.3        | 59% ± 1.5      | 68% ± 2.3      |
> |      | **Ours** | **73% ± 1.4**  | **55% ± 1.1**  | **56% ± 1.3**    | **69% ± 1.2**  | **74% ± 1.0**  |
> |------|----------|----------------|----------------|------------------|----------------|----------------|
> | 2021 | GA       | 58% ± 1.6      | 44% ± 1.3      | 45% ± 1.5        | 44% ± 1.4      | 68% ± 2.2      |
> |      | AL       | 61% ± 1.3      | 39% ± 1.2      | 45% ± 1.0        | 33% ± 1.5      | 66% ± 1.7      |
> |      | AML      | 77% ± 2.0      | 50% ± 1.5      | 50% ± 1.7        | 50% ± 1.4      | 86% ± 2.1      |
> |      | OML      | 54% ± 1.9      | 39% ± 1.4      | 45% ± 1.6        | 38% ± 1.7      | 59% ± 1.9      |
> |      | Prithvi  | 64% ± 1.7      | 48% ± 1.4      | 48% ± 1.3        | 48% ± 1.1      | 75% ± 1.8      |
> |      | UCB      | 72% ± 1.5      | 42% ± 1.2      | 40% ± 1.1        | 44% ± 1.3      | 87% ± 1.9      |
> |      | **Ours** | **80% ± 0.9**  | **53% ± 1.0**  | **58% ± 1.2**    | **53% ± 1.1**  | **95% ± 0.8**  |
> |------|----------|----------------|----------------|------------------|----------------|----------------|
> | 2019 | GA       | 57% ± 1.5      | 45% ± 1.3      | 49% ± 1.4        | 48% ± 1.7      | 60% ± 2.0      |
> |      | AL       | 66% ± 1.8      | 45% ± 1.5      | 45% ± 1.3        | 44% ± 1.6      | 77% ± 2.1      |
> |      | AML      | 74% ± 2.0      | 49% ± 1.4      | 49% ± 1.5        | 50% ± 1.3      | 86% ± 2.0      |
> |      | OML      | 66% ± 1.7      | 45% ± 1.2      | 46% ± 1.4        | 43% ± 1.5      | 77% ± 1.9      |
> |      | Prithvi  | 55% ± 1.6      | 36% ± 1.3      | 38% ± 1.4        | 34% ± 1.5      | 67% ± 1.8      |
> |      | UCB      | 42% ± 1.3      | 38% ± 1.1      | 49% ± 1.2        | 48% ± 1.4      | 39% ± 1.6      |
> |      | **Ours** | **74% ± 1.0**  | **49% ± 0.9**  | **49% ± 1.0**    | **50% ± 1.1**  | **86% ± 0.9**  |

---

> > ### Comment · Reviewer_wCqr · 2025-11-25
> > **Thanks to the authors for their response**
> >
> > I thank the authors for their response! In the revised paper, the caption of table 2 should clearly state what the +/- quantities are (standard deviations, I think from the author response). Adding standard deviations to tables 4,5,6 would be a benefit as well.

---

> ### Author Response · Authors · 2025-11-26
> **Acknowledgement**
>
> Dear Reviewer wCqr,
>
> We appreciate your acknowledgment and the additional feedback you provided. We have incorporated all your suggestions into the revised draft.
>
> Thanks,
>
> The Authors

---

### Official Review · Reviewer_j5o6 · 2025-11-01

**Soundness:** 3
**Presentation:** 1
**Contribution:** 2
**Rating:** 4
**Confidence:** 4

**Summary:**

This paper proposes a method for geospatial target discovery under tight resource constraints. The method combines multiple strategies such as relevance-uncertainty guided sampling and relevance-aware meta-batch formation for efficient decision-making. The experiments on searching for specific land cover types and contamination areas show that the method can generalize better than baselines to real-world scenarios under data constraints.

**Strengths:**

1. The problem of Open-World Learning for Geospatial Prediction and Sampling (OWL-GPS) is important and relevant to many real-world applications.
2. The method design addresses important challenges in the OWL-GPS problem, including strict sampling budgets and evolving data distributions, and non-revisitable inputs.
3. The experimental results show the improvements over the baselines.

**Weaknesses:**

1.  The contribution of the components are not well explained. Is the relevance uncertainty from domain-specific concepts like spectral channels trying to use selected features based on domain knowledge to do the sampling? This feels like a reasonable domain application choice but the novelty is not apparent.

2. Although the paper provides additional details on the datasets in the supplementary, it would be helpful to clarify how the characteristics of each dataset correspond to the OWL-GPS problem. For instance, Sentinel imagery is, in principle, abundantly available—so in what sense does it become “costly,” “non-replayable,” or “resource-constrained” in this work? Making this explicit for each dataset would also help readers understand how the baseline models are configured under the same setting. How difficult is it to get the labels?

3. The experimental setup lacks sufficient detail and makes it difficult to understand how the models were trained.

**Questions:**

Please explain the key technical contribution on uncertainty and sampling diversity compared with existing studies.

---

> ### Author Response · Authors · 2025-11-20
> **Response to Reviewer j5o6**
>
> We thank the reviewer for the thoughtful and constructive evaluation of our work. We appreciate the recognition of the importance of the OWL-GPS problem setting and the strengths highlighted in our formulation, method design, and empirical results. We are grateful for the reviewer’s insightful questions, which help us further clarify key aspects of our approach. We address each point below.
>
> > **Q1** Is the relevance uncertainty from domain-specific concepts like spectral channels trying to use selected features based on domain knowledge to do the sampling?
>
> **A1** The components of our framework serve distinct and complementary roles. The relevance encoder is **not** based on manually selected spectral channels or hand-crafted domain variables. Instead, we have a **pretrained ViT that operates directly on raw geospatial inputs** to produce a latent concept embedding. The “relevance uncertainty” comes from the **CVAE posterior over this latent space**, quantifying how confidently the model can characterize the semantic composition of a candidate region.
>
> Thus, the relevance signal does not depend on domain-specific feature engineering or expert-defined concept metadata. It is a data-driven, end-to-end–learned representation that is applicable across domains (PFAS, land cover, etc.). The novelty lies in introducing **a relevance-aware latent representation combined with its uncertainty through a CVAE**, and using this representation to guide sampling within the OWL-GPS setting.
>
>
> > **Q2** Sentinel imagery is, abundantly available—so in what sense does it become “costly,” “non-replayable,” or “resource-constrained” in this work? How difficult is it to get the labels?
>
> **A2** The OWL-GPS setting models constraints on supervision, not imagery. **Sentinel imagery is indeed abundant; however, the labels required for the discovery tasks in our datasets are not**. In real scientific monitoring problems like PFAS contamination, labeled examples are scarce, expensive, and slow to acquire. Moreover, OWL-GPS assumes an online, memory-limited regime in which the agent cannot store or replay past inputs beyond a small fixed buffer. These are the core constraints the framework is designed to capture.
>
> - PFAS (2019 & 2021): Each PFAS label corresponds to an actual field measurement collected by environmental agencies. Such **measurements require field deployment, sample collection, laboratory chemical analysis, and administrative coordination**. As a result, very few labeled samples exist, and expanding coverage is constrained by budget, personnel, and lab throughput. This is the source of “costly” and “resource-constrained.” In addition, under OWL-GPS assumptions, inputs are non-replayable: once a geospatial region has been processed and leaves the limited memory buffer, it cannot be revisited for training or queried again. The agent therefore must decide which regions to label as they arrive.
>
> - Land Cover (LC): For LC, labels are not inherently difficult to acquire, but **the purpose of including LC is to demonstrate that the framework generalizes beyond PFAS**. To place LC into a discovery setting comparable to PFAS, we **sparsify the supervision as described in the Appendix P**. This creates a regime where only a small subset of tiles are labeled, matching the limited-supervision assumption of OWL-GPS. We clarify that this sparsification is simulated to evaluate generalization, not a claim about real-world LC label difficulty.
>
> **We have made these distinction clear in the updated draft in Appendix section P**.
>
>  > **Q3** Explain the key technical contribution on uncertainty and sampling diversity compared with existing studies.
>
> **A3** Our approach advances beyond existing uncertainty and diversity-based sampling methods, such as CoreSet and BADGE, which rely on static, replayable unlabeled pools and conventional clustering techniques—settings incompatible with OWL-GPS’s streaming, non-replayable data and limited buffer constraints. Technically, **we model uncertainty in both label and latent relevance space via a CVAE, capturing not only epistemic uncertainty but also semantic novelty critical for geospatial discovery.** **For sampling diversity, our coverage-driven Greedy Intersection Algorithm (GIA) (Appendix J) clusters relevance vectors within the core buffer, constructing ε-hypercubes and selecting diverse meta-batches under strict streaming constraints. Importantly, standard approaches like DBSCAN (Appendix J, Table 11) underperform in this regime, underscoring the necessity of coverage-based methods.** Together, the relevance-aware uncertainty and GIA diversity mechanisms yield a technically distinct, streaming-compatible strategy that robustly supports OWL-GPS’s operational requirements.

---

> ### Author Response · Authors · 2025-11-20
> **Response to Reviewer j5o6**
>
> > **Q4** The experimental setup lacks detail
>
> **A4** OWL-GPS introduces a new constrained, streaming, non-replayable problem formulation, and **as reviewer wCqr noted, existing methods are not originally designed for this setting**. Therefore, each baseline must be carefully adapted to comply with the OWL-GPS constraints such as streaming inputs, no revisitation, limited memory and query budget, and updates only through queried labels. Below, we provide explicit descriptions of how every baseline is trained and evaluated under this unified setup.
>
> **Active Learning**: Successful performance in OWL-GPS requires a delicate balance between exploration and exploitation during inference. AL methods primarily focus on exploration by querying areas of highest model uncertainty. While this ensures thorough coverage, it overlooks exploiting accumulated knowledge about promising regions. Consequently, AL struggles to efficiently navigate the trade-off necessary for high performance in OWL-GPS. For training with the active learning baseline, we adhere strictly to the standard maximum uncertainty strategy for selecting samples to query during a standard supervised training with a ViT. For evaluation, we apply our sampling strategy with one key modification: the selection of queried samples is driven exclusively by the exploration score, with no consideration given to the exploitation score in the sampling process.
>
> **Greedy Approach**: GA represents the contrasting extreme by prioritizing exploitation — querying locations where the model predicts the highest likelihood of the target. Purely exploitation-driven strategies like GA risk premature convergence to suboptimal regions and fail to sufficiently explore the environment. Our experimental findings corroborate this limitation, showing inferior performance of GA relative to our method. GA conducts a single forward pass over all incoming regions during evaluation after a standard supervised training phase with a ViT, assigning confidence scores to each. It then selects the top-K regions based on the query budget. GA does not perform online updates, store past regions, or revisit inputs, making it a strict single-pass, non-adaptive baseline within OWL-GPS.
>
> **Prithvi**: Although Prithvi is a powerful framework, it is not specifically designed to handle the multi-faceted challenges of the OWL-GPS problem, such as continual adaptation in dynamic environments and robust online updating. These complex constraints underscore the need for a tailored approach like ours. Specifically, the geospatial foundation model Prithvi is trained using a standard supervised approach. During evaluation, it assigns confidence scores to unobserved samples, selects the highest scoring sample, updates its parameters with the latest observations, and repeats this process for subsequent batches until the query budget is exhausted.
>
> **UCB**: At first glance, the OWL-GPS problem may appear similar to a Multi-Armed Bandit scenario, making UCB a natural baseline. However, unlike typical MAB settings where each arm is independent, geospatial environments exhibit strong spatial correlations—observing one location informs us about neighboring regions. UCB does not leverage this spatial structure, treating each arm independently. This lack of spatial modeling fundamentally limits its performance in our problem setting, as reflected in Table 2 in our paper. We would like to emphasize that UCB is adapted to the OWL-GPS setting by treating each incoming region as an “arm” whose score is computed from its predicted reward (model confidence) and an exploration bonus. Once a region is queried based on the score, the model updates online using the new labeled sample, following the same online meta-update procedure as in our framework. UCB does not access any historical unlabeled pool, and all decisions are made per-arrival as in our framework.
>
> **Online Meta-Learning**: OML in our setup follows the same online meta-update rule used in our framework. The key difference is that OML does not use our meta-batch formation strategy with GIA for constructing diverse update sets. Instead, OML directly applies the update using the examples present in our buffer at that moment. Aside from the absence of GIA, OML operates identically to our framework under OWL-GPS constraints: streaming inputs, no revisiting beyond the small buffer, and continuous online meta-adaptation from the sparse labeled data encountered so far.
>
> **Active Meta-Learning**: AML follows a training approach similar to ours but differs in two key ways: (a) it forms the meta-training batch using standard random sampling from the buffer, and (b) it lacks an online update mechanism, unlike our proposed strategy. Our experiments demonstrate that AML is ineffective in the OWL-GPS setting, highlighting the critical role of an online update mechanism combined with a coverage-based meta-batch formation strategy.
>
> **We've included these details in Appendix U**.

---

### Official Review · Reviewer_Hb3E · 2025-11-01

**Soundness:** 2
**Presentation:** 3
**Contribution:** 3
**Rating:** 4
**Confidence:** 4

**Summary:**

This paper proposes a framework for policy-driven, online meta-learning under constrained sampling and memory settings in geospatial environments. It formalizes the OWL-GPS problem, which introduces strict assumptions: non-revisitable inputs, dynamically shifting distributions, and hard query budgets. To address this, the authors propose:
* A relevance-guided Conditional Variational Autoencoder (CVAE) to encode latent concept relevance.
* A relevance-aware sampling strategy that balances exploration and exploitation.
* A meta-training mechanism using core and reservoir buffers formed through Greedy Intersection Clustering (GIA) over relevance vectors.
* A theoretical justification of sampling criteria via entropy-based exploration (Theorem 1 and 2).
* Experiments on PFAS contamination and land cover tasks from satellite imagery data (years 2019 and 2021), showing superiority over baselines.

**Strengths:**

* The Greedy Intersection Algorithm for meta-batch diversity is clever and attempts to maintain semantic coverage without task boundaries.

* Ablation studies cover a range of hyperparameters and show some resilience of the method.

* The integration of domain-specific concepts into relevance encoding via CVAE is a principled and interpretable modeling choice.

**Weaknesses:**

* The experiments use only two datasets. Only one run per configuration is reported with no variance bars. SR  gains over some baselines are margin.

* Equation (6) uses a handcrafted combination of latent relevance distance and decoder uncertainty. The sampling objective is ad-hoc and insufficiently validated.

* The framework assumes domain experts can predefine concept variables that are always available. This is rarely feasible in real domains or multi-modal geospatial datasets.

* The SR formula includes a denominator term Ut defined as the maximum number of target pixels in the queried image. However, this is part of the ground truth and should not be used in evaluation.

**Questions:**

* All reported results appear to come from a single run. Did you evaluate across multiple random seeds? Can you report mean and standard deviation?
* Why was this form chosen in Equation (6)? Did you try simpler or standard acquisition functions like entropy or margin?
* How would your method adapt to domains without explicit concept metadata? Have you tested robustness to missing, noisy, or misaligned concepts? Can the concept encoder be replaced with a learned representation?

---

> ### Author Response · Authors · 2025-11-20
> **Response to Reviewer Hb3E**
>
> We thank the reviewer for the thorough and constructive feedback. We appreciate the detailed comments on the experimental setup, sampling formulation, and use of concepts, and we are glad that the reviewer found strengths in our meta-batch diversity strategy, ablation breadth, and principled modeling choices. We address each concern below.
>
> > **Q1.** Only two datasets?
>
> **A1** Our primary goal is to evaluate our framework within **real scientific discovery scenarios under the OWL-GPS setting, characterized by scarce datasets and costly, expert-dependent label collection.** The PFAS case exemplifies this, with two datasets from 2019 and 2021 showcasing **distinct spatiotemporal distributions**, providing diverse evaluation contexts. **To assess technical generalizability, we also include a third dataset (LandCover (LC)) from a different sensing modality.** Since standard LC segmentation benchmarks do not reflect the sparse-discovery constraints relevant to OWL-GPS, we developed a **sparse variant (detailed in Appendix P**) to better simulate realistic conditions.
>
> > **Q2.** SR gains are marginal?
>
> **A2** While some classification metrics show similar values across baselines, the SR metric—which captures the number of distinct target-rich regions discovered under strict sampling budgets—exhibits **substantial and consistent improvements**. For instance, on the most challenging PFAS-2021 dataset, OWL-GPS attains an **SR of 0.95**, significantly outperforming **UCB (0.87), Prithvi (0.75), AML (0.86), Greedy (0.68), AL (0.66), and OML (0.59)**, as detailed in Table 2. Comparable performance gaps also appear on PFAS-2019 and the sparse LandCover datasets. These gains are far from marginal—they translate to discovering markedly more distinct target regions with the same number of samples, aligning precisely with the OWL-GPS operational objective. **Even modest percentage improvements yield meaningful real-world impact, where each additional discovered region carries significant environmental or operational value**.
>
>
> > **Q3** Why this form chosen in Eqn. (6)? Did you try simpler or standard acquisition functions like entropy?
>
> **A3** The form of Equation (6) directly embodies the two essential signals driving effective discovery in OWL-GPS:
> * (1) **Latent relevance distance**, which distinguishes candidate regions by their concept composition divergence from previously sampled ones, and
> * (2) **Decoder uncertainty**, capturing epistemic uncertainty in predictions.
>
> Our theoretical analysis (**Appendix H, Thm 1)** rigorously justifies that maximizing entropy under model assumptions leads naturally to this combined selection criterion, coupling distance between latent means and predictive variance.
> Empirically, we validated this approach by comparing it against classical acquisition functions—**such as UCB (Table 2)**—and **conducting an ablation where we removed the relevance-distance term (Table 6)**, reducing Eq. (6) to uncertainty-only sampling.
> **Both alternatives underperformed relative to our method, confirming that integrating exploration in relevance space with decoder uncertainty is crucial for robust active geospatial search in this framework.**
>
> > **Q4** How to adapt to domains without explicit concept metadata? Have you tested robustness to missing/noisy concepts? Can concept encoder be replaced with a learned representation?
>
> **A4** Our framework leverages domain-informed concept variables when available, such as land cover and facility proximity. **To test robustness to incomplete or noisy concepts, we performed a concept-masking experiment (Appendix F, Table 9), randomly dropping concept dimensions at inference. The model retains strong performance, with SR only decreasing from 95% to 88%, demonstrating resilience to missing or noisy concept data.**
>
> Importantly, our "concept encoder" is a pretrained Vision Transformer that learns latent concept embeddings directly from raw geospatial inputs, not relying on manually specified metadata at sampling time.
>
> > **Q5** Justify the role of Ut in SR metric.
>
> **A5** The term Ut appears solely in the evaluation metric and is not utilized during the sampling or decision-making phases of the method. It serves to normalize the success Rate for each queried region, ensuring consistent evaluation across regions that vary in size and target density. Importantly, this normalization is applied post hoc—after queries are made—which precludes any information leakage to the model. Employing ground-truth information (such as IoU or precision/recall) for evaluation is a standard practice that does not compromise model integrity.

---

> > ### Author Response · Authors · 2025-11-20
> > **Response to Reviewer Hb3E**
> >
> > > **Q6** Did you evaluate across multiple random seeds? Can you report mean and standard deviation?
> >
> >  All reported results are averaged over three independent random seeds. We apologize for not mentioning this earlier in the paper.
> >  Below, we provide the updated main results table with the corresponding variance bars. We have included these variance bars in the revised draft.
> >
> > ### Table 2: Comparison with Baselines (Mean ± Std over 3 seeds)
> >
> > | Year | Method  | Acc.           | F-score        | Precision       | Recall         | SR             |
> > |------|----------|----------------|----------------|------------------|----------------|----------------|
> > | LC   | GA       | 46% ± 1.2      | 35% ± 1.0      | 48% ± 0.9        | 40% ± 1.3      | 47% ± 1.4      |
> > |      | AL       | 58% ± 1.8      | 41% ± 1.5      | 50% ± 1.2        | 47% ± 1.9      | 60% ± 2.0      |
> > |      | AML      | 53% ± 2.0      | 40% ± 1.3      | 65% ± 1.6        | 30% ± 1.8      | 54% ± 2.1      |
> > |      | OML      | 67% ± 1.7      | 49% ± 1.6      | 82% ± 1.8        | 39% ± 1.9      | 67% ± 1.5      |
> > |      | Prithvi  | 61% ± 2.0      | 40% ± 1.4      | 51% ± 1.5        | 56% ± 1.7      | 61% ± 1.8      |
> > |      | UCB      | 67% ± 2.1      | 48% ± 1.7      | 53% ± 1.3        | 59% ± 1.5      | 68% ± 2.3      |
> > |      | **Ours** | **73% ± 1.4**  | **55% ± 1.1**  | **56% ± 1.3**    | **69% ± 1.2**  | **74% ± 1.0**  |
> > |------|----------|----------------|----------------|------------------|----------------|----------------|
> > | 2021 | GA       | 58% ± 1.6      | 44% ± 1.3      | 45% ± 1.5        | 44% ± 1.4      | 68% ± 2.2      |
> > |      | AL       | 61% ± 1.3      | 39% ± 1.2      | 45% ± 1.0        | 33% ± 1.5      | 66% ± 1.7      |
> > |      | AML      | 77% ± 2.0      | 50% ± 1.5      | 50% ± 1.7        | 50% ± 1.4      | 86% ± 2.1      |
> > |      | OML      | 54% ± 1.9      | 39% ± 1.4      | 45% ± 1.6        | 38% ± 1.7      | 59% ± 1.9      |
> > |      | Prithvi  | 64% ± 1.7      | 48% ± 1.4      | 48% ± 1.3        | 48% ± 1.1      | 75% ± 1.8      |
> > |      | UCB      | 72% ± 1.5      | 42% ± 1.2      | 40% ± 1.1        | 44% ± 1.3      | 87% ± 1.9      |
> > |      | **Ours** | **80% ± 0.9**  | **53% ± 1.0**  | **58% ± 1.2**    | **53% ± 1.1**  | **95% ± 0.8**  |
> > |------|----------|----------------|----------------|------------------|----------------|----------------|
> > | 2019 | GA       | 57% ± 1.5      | 45% ± 1.3      | 49% ± 1.4        | 48% ± 1.7      | 60% ± 2.0      |
> > |      | AL       | 66% ± 1.8      | 45% ± 1.5      | 45% ± 1.3        | 44% ± 1.6      | 77% ± 2.1      |
> > |      | AML      | 74% ± 2.0      | 49% ± 1.4      | 49% ± 1.5        | 50% ± 1.3      | 86% ± 2.0      |
> > |      | OML      | 66% ± 1.7      | 45% ± 1.2      | 46% ± 1.4        | 43% ± 1.5      | 77% ± 1.9      |
> > |      | Prithvi  | 55% ± 1.6      | 36% ± 1.3      | 38% ± 1.4        | 34% ± 1.5      | 67% ± 1.8      |
> > |      | UCB      | 42% ± 1.3      | 38% ± 1.1      | 49% ± 1.2        | 48% ± 1.4      | 39% ± 1.6      |
> > |      | **Ours** | **74% ± 1.0**  | **49% ± 0.9**  | **49% ± 1.0**    | **50% ± 1.1**  | **86% ± 0.9**  |

---

### Author Response · Authors · 2025-11-26
**Follow-up**

Dear Reviewers,

We would like to again thank the reviewers for their insightful feedback, which has greatly helped us strengthen our analysis. As we approach the final week of the discussion period, we would like to ensure that the reviewers’ concerns have been fully addressed. If there are any remaining points or questions, we would be very happy to engage further in discussion to provide additional clarification.

Thanks,

The Authors

---

### Author Response · Authors · 2025-12-02
**Response Summary and a Final Note to AC**

Dear Area Chair,

We sincerely thank the reviewers for their thoughtful and constructive feedback. Below, we provide a summary of the response:

* In revising the manuscript, we clarified several methodological details, expanded key descriptions, and incorporated additional complementary analyses suggested by the reviewers to further improve clarity. We clarified dataset choices and strengthened the generalization discussion with the (already included) LandCover dataset and its sparse-supervision variant.


* We also more prominently highlighted the existing ablation studies in the appendix (relevance encoder, GS orthogonalization, uncertainty terms, and GIA diversity), which were previously overlooked by the reviewers in their initial reviews, and we added variance bars to all reported results to better convey variability.


* We also further explain the acquisition function (Eq. 6), the role of pixel-level supervision, the robustness of our approach to missing or noisy concepts, and the adaptation of baselines to OWL-GPS’s streaming, non-replayable setting.


* To support interpretability, we added concise visualizations depicting concept-relevance patterns (comparing correct vs. incorrect samples), context-adaptive relevance across geographic regions, and sampling-trajectory plots that demonstrate the transition from exploration to exploitation.


* Finally, we expanded the related-work discussion, included explicit time and memory complexity details, and addressed all minor editorial issues in the revised draft.


We hope these revisions adequately address the reviewers’ concerns and appreciably strengthen the manuscript.

We kindly request you to take into consideration our detailed point-by-point responses, which address concerns raised by reviewers who were unable to respond fully due to the discussion period being cut short. We sincerely appreciate your effort and time in overseeing the review process.


Thanks,

The Authors

---

### Note · Program_Chairs · 2026-01-17
**Submission Desk Rejected by Program Chairs**

The following references in this submission do not refer to real documents and/or have major errors in bibliographic information:

 Yifei Zheng, Haoran Zhang, Satinder Singh, Michael Kearns, Aaron Roth, and Zhuoran Yang. Efficient exploration in pomdps with limited memory. In International Conference on Machine Learning (ICML), pp. 27207-27236. PMLR, 2022.